# BENCHMARKING LLMS' JUDGMENTS WITH NO GOLD STANDARD

**Shengwei Xu**[*][†]
School of Information
University of Michigan, Ann Arbor, USA
shengwei@umich.edu

**Yuxuan Lu**[*][‡]
School of Computer Science
Peking University, Beijing, China
yx_lu@pku.edu.cn

**Grant Schoenebeck**[†]
School of Information
University of Michigan, Ann Arbor, USA
schoeneb@umich.edu

**Yuqing Kong**[‡]
School of Computer Science
Peking University, Beijing, China
yuqing.kong@pku.edu.cn

## ABSTRACT

We introduce the GEM (Generative Estimator for Mutual Information), an evaluation metric for assessing language generation by Large Language Models (LLMs), particularly in generating informative judgments, without the need for a gold standard reference. GEM broadens the scenarios where we can benchmark LLM generation performance-from traditional ones, like machine translation and summarization, where gold standard references are readily available, to subjective tasks without clear gold standards, such as academic peer review.

GEM uses a generative model to estimate mutual information between candidate and reference responses, without requiring the reference to be a gold standard. In experiments on a human-annotated dataset, GEM demonstrates competitive correlations with human scores compared to the state-of-the-art GPT-4o Examiner, and outperforms all other baselines. Additionally, GEM is more robust against strategic manipulations, such as rephrasing or elongation, which can artificially inflate scores under a GPT-4o Examiner.

We also present GRE-bench (Generating Review Evaluation Benchmark) which evaluates LLMs based on how well they can generate high-quality peer reviews for academic research papers. Because GRE-bench is based upon GEM, it inherits its robustness properties. Additionally, GRE-bench circumvents data contamination problems (or data leakage) by using the continuous influx of new open-access research papers and peer reviews each year. We show GRE-bench results of various popular LLMs on their peer review capabilities using the ICLR2023 dataset.

## 1 INTRODUCTION

High-quality and reliable Large Language Model (LLM) benchmarks can effectively guide research, encourage innovation, monitor their advancement, and inform users of which model to use for their purpose. The importance of the final goal is underscored by the over 900k models currently available on Hugging Face, an online platform for open-source LLMs[1]. Various benchmarks are proposed for evaluating LLMs' ability in different aspects, including ARC (Chollet, 2019), HellaSwag (Zellers et al., 2019), Massive Multitask Language Understanding (MMLU) (Hendrycks et al., 2020), GSM8K (Cobbe et al., 2021), TruthfulQA (Lin et al., 2021), Natural Questions (Kwiatkowski et al., 2019), etc.

Most benchmarks are based on multiple-choice questions or other questions with objective gold standard answers, since it is easy to verify the LLMs' outputs. While they provide valuable evaluation for

---

[*]Both authors contributed equally to the paper.
[†]Supported by United States National Science Foundation award number 2313137.
[‡]Supported by National Natural Science Foundation of China award number 62372007.

[1]https://huggingface.co/docs/hub/en/index

LLMs, open-ended tasks, e.g. providing judgment about a research paper, encompass a broader array of skills and require both objective and subjective reasoning. In addition, concern has been raised about data contamination (also called data leakage), where the training data contains information about the tasks in the benchmarks, as the LLMs are pre-trained on massive online data, which is also the source of some of the benchmark tasks. LLMs can show unreliably good performance due to data contamination (Sainz et al., 2023; Golchin & Surdeanu, 2023; Oren et al., 2023). In contrast, with open-ended questions, LLMs can be asked to provide judgments about newly created content, e.g. the latest academic papers, for which LLMs have yet to index human evaluations or responses.

However, it is not clear how to automate the evaluation of subjective response quality. An added challenge is that there is no gold standard quality response with which to compare.

We would like an evaluator to have two properties. First, it should be **accurate** and be sensitive to the semantic content response. As current LLMs have already shown strong language ability, we want to focus on evaluating the semantic informativeness of candidate responses instead of their syntax or style. Second, the evaluator should be **manipulation-resistant**-we should not be able to manipulate a response in a trivial fashion to consistently increase evaluation scores. This is important because otherwise one cannot determine whether a high evaluation score indicates that the LLM output a high-quality response or merely results from manipulation designed to achieve an artificially inflated score.

Given these gaps in previous research, our research question is: **Can we develop accurate, manipulation-resistant, and automated evaluation metrics for textual responses with no gold standard reference to compare with?**

A straightforward approach may involve utilizing an alternative LLM as an oracle examiner to directly provide the evaluation, which has demonstrated efficacy in evaluating open-response QA (Bai et al., 2024), chatbot (Zheng et al., 2023), etc. However, LLM examiners can have blind spots and be susceptible to certain manipulations (Hada et al., 2023; Doddapaneni et al., 2024). In our experiments, elongating all responses by adding the same fixed sentences can significantly increase the GPT-4o LM examiner's score. Interestingly, previous research has shown that even human evaluations can be manipulated by adding meaningless text (Goldberg et al., 2023). Meanwhile, other automated natural language generation (NLG) evaluation metrics, including BLEU (Papineni et al., 2002), ROUGE (Lin, 2004), BERTScore (Zhang et al., 2019), BARTScore (Yuan et al., 2021) and GPTScore (Fu et al., 2023), rely on comparison with a gold standard reference.

**Our Contributions.** We propose the *Generative Estimator for Mutual information (GEM)* which uses the estimated Shannon mutual information (MI) between a set of candidate responses and a set of peer reference responses, which need not be of gold standard quality. The mutual information measures the amount of information revealed about the reference responses by obtaining the candidate responses. By artificially removing stylistic and syntactical information, our approach measures how much semantic information the candidate responses can reveal about the reference responses, which is related to the concept of *semantic coverage* (Yuan et al., 2021; Nenkova & Passonneau, 2004). We additionally propose a variant of our method, *Generative Estimator for Mutual Information with Synopsis (GEM-S)*, which estimates the mutual information conditional on a synopsis of the task, e.g. the abstract of the paper. This prevents a candidate response from receiving a high score based solely on superficial information. Consequently, the score emphasizes the *additional* semantic information gained from the candidate responses.

For implementation, we utilize a generative language model to estimate the conditional distribution between two text responses. Thus, the GEM can be categorized as an LLM-probability-based metric, with techniques similar to BARTScore (Yuan et al., 2021) and GPTScore (Fu et al., 2023), from which our metric inherits the effectiveness of evaluating objective tasks with a gold standard.

Our work establishes a bridge between the information theoretical framework in the literature of information elicitation without verification (Kong & Schoenebeck, 2019; Lu et al., 2024) and the NLG evaluation problem. Though GEM and GEM-S resemble the GPPM and GSPPM mechanisms in Lu et al. (2024), with manipulation resistance aligned to their incentive compatibility, we make a necessary change to make the score more suitable for the NLG evaluation problem. Specifically, for incentive compatibility, the score can rely solely on accuracy of predicting the peer reference, while we use mutual information to capture the gain in that accuracy for evaluation purposes.

**Results Overview.** The results of experiments validate GEM's accuracy and resistance to manipulation, and compare it with many different NLG evaluation metrics.

1. **Positive Correlation with Human Annotation** (Section 4.1) On a human-annotated dataset, the GEM metrics, especially GEM-S, achieve a significant positive correlation with human-labeled quality scores and demonstrate competitive performance with GPT4-o Examiner while outperforming all other baseline metrics.

2. **Better Sensitivity to Degradations** (Section 4.2) Compared to various baseline metrics, GEM and GEM-S are the only metrics that demonstrate significant sensitivity to all semantic degradations in our experiment, by effectively penalizing degraded responses.

3. **Better Robustness against Manipulations** (Section 4.3) GEM and GEM-S are the only metrics that exhibit no significant score increases after meaningless elongation (Figure 3) and GPT-4o/Llama-3.1 rephrase, whereas LMExaminer show vulnerabilities by significantly increasing scores after all manipulations.

Building on GEM and GEM-S, we introduce the **GRE-bench** (Generating Review Evaluation Benchmark) to evaluate LLMs' peer review capabilities with data from open-access research papers and peer reviews. With the continuous influx of new data each year, GRE-bench can mitigate data contamination issues. In addition, since GRE-bench is based on GEM or GEM-S, it inherits their nice properties demonstrated in the experiments above.

We run GRE-bench for various state-of-the-art LLMs on the ICLR 2023 dataset, and present the results in Section 5. We find a strong correlation between parameter size and GRE-bench score within LLM families, which further validates the effectiveness of GEM and GEM-S.

## 1.1 RELATED WORK

**LLMs in academic peer review.** Given the success of LLMs, they have the potential to assist with peer reviews when used appropriately (Robertson, 2023; Kuznetsov et al., 2024). Liang et al. (2023) use a survey to evaluate GPT-4-generated reviews, and find that the GPT-generated reviews are thought helpful by more than 50% of participants. However, there have been concerns regarding issues such as hallucinations (Donker, 2023) and inconsistent performance Liu & Shah (2023). While our study uses the peer review scenario to evaluate LLM capabilities, it also provides a quantitative comparison for the effectiveness of various LLMs in generating informative reviews.

**Information Elicitation.** Information elicitation employ a principal-agent model (Ali & Silvey, 1966) where the principal aims to elicit information from the agent, e.g. the probability of rain tomorrow. To incentivize the agent to provide truthful and informative reports, payment mechanisms are employed that reward truthful and informative reports more than untruthful or uninformative ones. When the information is verifiable, e.g., we will eventually know whether it rains tomorrow, *proper scoring rules* (Good, 1952; Brier, 1950; Hendrickson & Buehler, 1971) can be applied, which are similar to the loss functions in supervised learning. When the information is subjective and unverifiable, Miller et al. (2005) propose the *peer prediction* mechanism, suggesting rewarding data by how well it predicts a peer's subjective report which is judged by a proper scoring rule. Prelec (2004) propose the Bayesian Truth Serum mechanism. Kong & Schoenebeck (2019) propose an information-theoretic framework, suggesting paying the agents according to the f-mutual information between their reports. The framework provides a unified theoretical view of previous mechanisms in Dasgupta & Ghosh (2013), and Prelec (2004). Lu et al. (2024) first generalize the peer prediction to elicit informative textual responses with LLMs, providing the foundation for our GEM metrics.

## 2 PRELIMINARIES

**Model.** We consider a scenario with a **candidate** under evaluation and a peer **reference** working on the same tasks. We denote $\mathcal{W}$ as the space of all tasks, with an underlying distribution over it, denoted as $\Delta\mathcal{W}$. Given a sample task $W \sim \Delta\mathcal{W}$, both the candidate and the reference generate a response, represented by the random variable $X$ and $Y$ respectively. The response space for both $X$ and $Y$ is denoted by $\Sigma$. The tuple $(W,X,Y)$ follows an underlying distribution $\mathcal{D} := \Delta(\mathcal{W} \times \Sigma^2)$. *Note that the reference's responses do not need to be gold standard.* We assume that $X$ and $Y$ are independent conditioning on the task $W$.

Empirically, in a dataset $D$, we have a list of sampled tasks $\mathbf{w} = \{w_1, \cdots, w_n\}$, and a list of corresponding reference responses $\mathbf{y} = \{y_1, \cdots, y_n\}$. Let the candidate generate a list of responses $\mathbf{x} = \{x_1, \cdots, x_n\}$ to the corresponding tasks. We adopt a common assumption that all the samples $(w_i, x_i, y_i)$ are generated i.i.d. following the underlying distribution $\mathcal{D}$.

**Benchmarking the candidate.** The *evaluation metric* $f : \mathcal{W} \times \Sigma^2 \to \mathbb{R}$ maps a tuple $(w, x, y)$ to a *score $S$*. Note that some evaluation metrics only utilize one or two of the items in the tuple. If the evaluation metric utilizes a language model to compute the score, we call the model used as the *evaluation-LM*. Different evaluation-LMs may lead to different evaluation results. A nice property of our GRE-bench is the consistency over evaluation-LMs, i.e., using a smaller evaluation-LM will not significantly change the evaluation results, we will discuss it in Appendix A.2. Given a dataset $D$ and an evaluation metric $f$, we can estimate $\mathbb{E}_{(W,X,Y) \sim \mathcal{D}}[f(W, X, Y)]$, by using the average of the evaluation metrics over all samples, which is named the *benchmark*.

**Shannon Mutual Information.** Ideally, we aim to score the candidate according to the informativeness of her responses, thus, Shannon Mutual Information (MI) (Shannon, 1948) is a suitable metric. The MI between two random variables, $X$ and $Y$, provides a quantitative measure of the information shared between them. Specifically, it assesses how much knowing one variable can inform us about the other. This is formally defined as: $\mathrm{I}(X;Y) = \sum_{x,y} \Pr[X = x, Y = y] \log \frac{\Pr[X=x, Y=y]}{\Pr[X=x]\Pr[Y=y]}$. The conditional mutual information $I(X; Y \mid Z)$ measures the mutual information between $X$ and $Y$, conditioned on a random variable $Z$: $\mathrm{I}(X;Y|Z) = \sum_z \Pr[Z = z]\mathrm{I}(X;Y|Z = z)$.

In an academic peer review scenario, $Z$ can be some superficial information of a paper (e.g. the abstract), the conditional MI measures how much information is revealed beyond the abstract and emphasizes the MI gained from the semantics judgment.

**Point-wise Mutual Information as an Estimator.** To estimate the MI with samples, we employ Point-wise Mutual Information (PMI) (Fano & Hawkins, 1961; Church & Hanks, 1990), an unbiased estimator of the MI. Formally, the PMI between realization $x$ and $y$ is defined as

$$\mathrm{PMI}(x;y) := \log \frac{\Pr[X = x, Y = y]}{\Pr[X = x]\Pr[Y = y]} = \log\Pr[Y = y \mid X = x] - \log\Pr[Y = y].$$

Similarly, the PMI between $x$ and $y$ conditional on $z$ is defined as

$$\mathrm{PMI}(x;y \mid z) = \log\Pr[Y = y \mid X = x, Z = z] - \log\Pr[Y = y \mid Z = z].$$

## 3 GENERATIVE ESTIMATOR FOR MUTUAL INFORMATION (GEM)

In this section, we propose the Generative Estimator for Mutual Information (GEM), prove its theoretical effectiveness even if the reference's responses are not gold standard under certain assumptions, and discuss its empirical practice.

As discussed in Section 2, we use the mean of sample PMIs as an unbiased estimator of the MI. Given a sampled tuple with a task $w_i$, a candidate response $x_i$ and a reference response $y_i$, we aim to compute $\mathrm{PMI}(x_i; y_i) = \log\Pr[Y = y \mid X = x] - \log\Pr[Y = y]$.

Following Yuan et al. (2021); Fu et al. (2023); Lu et al. (2024), we use a generative language model to estimate the probability distribution of the reference response conditional on the candidate's output, denoted by $\Pr_{\mathrm{LLM}}[Y = y \mid X = x]$, and the marginal distribution, denoted by $\Pr_{\mathrm{LLM}}[Y = y]$.

Specifically, we use the LLMs' inherent token prediction function. We tokenize $y$ as $y = y^{(1)} y^{(2)} \cdots y^{(|y|)}$, and then integrate $x$ into the prompt. Given any length-$k$ prefix $y^{(1)} y^{(2)} \cdots y^{(k)}$, we use the LLM to predict $\Pr_{\mathrm{LLM}}[Y^{(k+1)} = y^{(k+1)} \mid Y^{(1)} \cdots Y^{(k)} = y^{(1)} \cdots y^{(k)}, X = x]$. With Bayesian rule, by multiplying over $k$, we have

$$\Pr_{\mathrm{LLM}}[Y = y \mid X = x] = \prod_{k=0}^{|y|} \Pr_{\mathrm{LLM}}[Y^{(k+1)} = y^{(k+1)} \mid Y^{(1)} \cdots Y^{(k)} = y^{(1)} \cdots y^{(k)}, X = x].$$

Similarly, we can estimate the marginal distribution $\log\Pr_{\mathrm{LLM}}[Y = y]$. Thus, we have the estimated PMI, denoted as $\widehat{\mathrm{PMI}}(x_i; y_i) = \log\Pr_{\mathrm{LLM}}[Y = y \mid X = x] - \log\Pr_{\mathrm{LLM}}[Y = y]$, and consequently, the estimated MI over dataset $D$ is $\widehat{I}(X;Y) = \frac{1}{n}\sum_{i=1}^n \widehat{\mathrm{PMI}}(x_i; y_i)$.

### 3.1 THEORETICAL GUARANTEES: BENCHMARKING LLMS WITH NO GOLD STANDARD

We now show that even when the reference is not gold standard, the estimated MI can still be a benchmark where better candidates achieve higher scores theoretically. We adopt a widely-used model in decision-making theory (Blackwell, 1951; 1953). The candidate's response follows an *information structure*, a mapping $\sigma : \mathcal{W} \to \Delta\Sigma$, where $\Sigma$ is the response space. It represents the distribution of responses conditional on the task.

Within this model, Blackwell's order (Blackwell, 1951; 1953) provides a partial order of the informativeness of information structures. Consider two candidates, H and L, whose responses $X_H$ and $X_L$ follow information structures $\sigma_H$ and $\sigma_L$ respectively. Information structure $\sigma_H$ is more informative than $\sigma_L$ in the sense of Blackwell order if there exists a stochastic mapping $\Gamma$, such that $\sigma_L = \Gamma\sigma_H$. Intuitively, the information structure with a lower Blackwell order is more noisy, and thus, can be regarded as lower quality.

We now show that higher Blackwell order information leads to (approximately) higher GEM scores when the LLM can provide a "good" estimation. Formally, we have

**Proposition 3.1.** *When the KL-divergence[2] between the LLM estimated distribution and the underlying distribution satisfies*

$$D_{KL}\Big[\Pr_{\text{LLM}}[Y=\cdot\,|\,X=x_H]\Big\|\Pr[Y=\cdot\,|\,X_H=x_H]\Big] < \epsilon, \forall x_H.$$

*For the two candidates H and L discussed above, the information structure of H Blackwell dominates L's, when the size of dataset n goes to infinity, we have $\widehat{I}(X_H;Y) \geq \widehat{I}(X_L;Y) - \epsilon$.*

The proof is provided in Appendix B, following previous information elicitation literature (Lu et al., 2024). It mainly relies on the data processing inequality (Shannon, 1948; Thomas & Joy, 2006). Additionally, we provide a similar proposition for conditional MI in Appendix B. We highlight that while this theoretical result requires a strong assumption, it gives intuitions of why and how our GEM scores could work. Real-data experiments are then conducted to further validate the effectiveness of GEM.

### 3.2 FILTERING OUT SHORTCUTS

Textual data is inherently high-dimensional, encompassing various aspects. For example, peer reviews may have a hierarchical information structure, including language style, the topic of the paper, and critical judgments, as shown in Figure 1. While each of these dimensions influences LLM predictions and consequently contributes to the mutual information, they are not equally important when assessing the quality of LLM-generated responses, particularly in the context of peer reviews.

Empirically, the correlation between superficial information can act as "shortcuts" (Geirhos et al., 2020) confounding LLM predictions, cause unintended bias in the estimated mutual information, and consequently distort the evaluation metric towards superficial correlations rather than thorough judgments. Since we focus on the semantic quality of these responses, we aim to filter out dimensions like language styles that could skew the evaluation metric by offering "shortcuts" in LLM predictions.

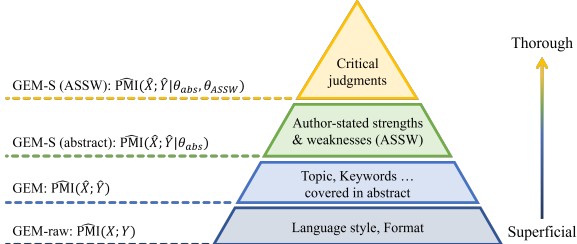

Figure 1: Example of Hierarchical Information Structure in Peer Reviews

**Preprocessing.** A straightforward yet effective method proposed by Lu et al. (2024), is preprocessing responses using a specific LLM to rephrase and summarize the content. This preprocessing step stan-

---

[2]The KL-divergence between two distributions over the same probability space is $D_{\text{KL}}(P\|Q) = \sum_x P(x)\log(P(x)/Q(x))$.

dardizes language style and eliminates superficial information, thereby reducing the influence of irrelevant dimensions. We denote the preprocessed versions of responses $x_i$ and $y_i$ as $\hat{x}_i$ and $\hat{y}_i$ respectively.

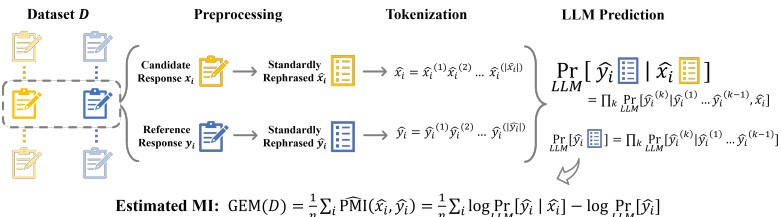

$$\text{Estimated MI: } \operatorname{GEM}(D) = \frac{1}{n}\sum_i \widehat{\operatorname{PMI}}(\hat{x}_i, \hat{y}_i) = \frac{1}{n}\sum_i \log \Pr_{LLM}[\hat{y}_i \mid \hat{x}_i] - \log \Pr_{LLM}[\hat{y}_i]$$

Figure 2: An Overview of Our Generative Estimator for Mutual Information

We have now outlined all steps of our Generative Estimator for Mutual Information (GEM) framework, as illustrated in Figure 2. The **GEM** metric between a candidate response $x_i$ and a reference response $y_i$ is defined as follows:

$$\operatorname{GEM}(x_i, y_i) := \widehat{\operatorname{PMI}}(\hat{x}_i, \hat{y}_i).$$

By taking the arithmetic average of all $\operatorname{GEM}(x_i, y_i)$ over the dataset $D$, we have an estimated mutual information between random variables $X$ and $Y$, as shown in Figure 2.

For comparison, we also introduce the **GEM-raw** metric calculated without preprocessing as follows: $\operatorname{GEM-raw}(x_i, y_i) := \widehat{\operatorname{PMI}}(x_i, y_i)$, which serves as a baseline.

**GEM-S: conditioning out the synopsis.** Kong & Schoenebeck (2018) suggest surface-level information can inflate mutual information through simple correlations, reducing the mutual information's effectiveness in reflecting semantic informativeness, especially when the responses have a hierarchical information structure as shown in Figure 1. As some superficial information can be contained in a synopsis of the task, e.g. the abstract of the paper to be reviewed, conditional mutual information can be used to distill the semantic informativeness (Lu et al., 2024).

The GEM-S metric is designed to mitigate this issue by conditioning out the synopsis of the task, denoted as $\theta(w_i)$. The **GEM-S** metric is formally defined as:

$$\operatorname{GEM-S}(\theta(w_i), x_i, y_i) = \widehat{\operatorname{PMI}}(\hat{x}_i, \hat{y}_i \mid \theta(w_i)).$$

With the conditional formulation above, only the additional, non-trivial mutual information is measured, thereby offering a more accurate assessment of the candidate's performance in providing information beyond superficial content.

Moreover, we can adjust the granularity of the synopsis to make the GEM-S metric focus on evaluating different aspects of the candidate LLMs' ability. In the example of Figure 1, the standard GEM measures the general ability to produce relevant reviews. By conditioning on the abstract, GEM-S (abstract) evaluates the LLM's ability to generate judgments, including retrieving author-stated strengths and weaknesses (ASSW) and providing critical feedback. By further conditioning on a summary of these author-stated strengths and weaknesses, GEM-S (ASSW) narrows the focus to the LLM's capacity for critical thinking and delivering constructive feedback.

## 4 EXPERIMENT: VALIDATING GEM'S EFFECTIVENESS

In this section, we provide our experiment setup to validate the effectiveness of the GEM metric in benchmarking LLMs without gold-standard references, and show the results. Code repository is at `https://github.com/yx-lu/Benchmarking-LLMs--Judgments-with-No-Gold-Standard`.

**GEM setup.** We present the empirical setup of the GEM metric and its variants. In the validation experiment in this section, Llama-3.1 8B is used as the evaluation-LM to estimate $\log\Pr_{\text{LLM}}[Y = y \mid X = x]$ and $\log\Pr_{\text{LLM}}[Y = y]$. After confirming the effectiveness of GEM metrics using the small model, for even better performance, we scaled up to a larger, 70B-parameter version of Llama-3.1 for computing the GRE-bench on the ICLR 2023 dataset in Section 5. We show the correlation between results of the smaller (8B) and larger (70B) models in Appendix A.2, to ensure robustness of our approach.

For text preprocessing, we employ GPT-4o. We also present the results with Llama-3.2 90B preprocessing in Appendix A.3, and demonstrate the robustness of GEM metrics with various preprocessing models. For GEM-S, without further indication, we use the abstract of the paper as the synopsis in this section. The prompts for probability estimation and preprocessing can be found in Appendix C.

**Baselines.** We compare the performance of GEM and its variants with the following established evaluation metrics commonly used in NLG evaluation, including two metrics from the pre-LLM era, *BLEU* (Papineni et al., 2002) and *ROUGE-L* (Lin, 2004), and an embedding-based metric, *BERTScore* (Zhang et al., 2019), a probability-based metric *BARTScore* (Yuan et al., 2021), and a GPT-based metric *LMExaminer (Language Model as Examiner)* (Bai et al., 2024). The detailed implementation of all baselines is provided in Appendix D.1.

## 4.1 RESULT 1: POSITIVE CORRELATION WITH HUMAN ANNOTATION

In this section, we show the experiment results utilizing a human-annotated dataset to validate how well the GEM metrics correlate with human-labeled quality scores compared to various baselines. A higher correlation indicates that the evaluation metric better aligns with human preference.

Note that, a perfect correlation can be impossible to achieve for several reasons: Human annotations are inherently noisy (Goldberg et al., 2023), evaluation of individual responses can also be noisy due to subjectivity, and there is no gold-standard reference. Nonetheless, our goal is to demonstrate systematic-level positive correlation.

**Score.** For task $w_i$ with corresponding responses $\mathbf{x}_i = \{x_{ij}\}$ where $x_{ij}$ is the $j$-th response. The score of response $x_{ij}$ is computed by $\frac{1}{|\mathbf{x}_i|-1}\sum_{k \neq j} f(w_i, x_{ij}, x_{ik})$ where $f$ is the evaluation metric.

**Human-Annotated Peer Grading Dataset.** This dataset contains 30 student project proposals in a graduate-level class on machine learning and about 180 peer evaluations of the proposals. We use GPT-4o to generate a short abstract for each proposal. Each proposal has at least 4 reviews, and each review contains an overall score and textual feedback, including "Strengths of the project", "Weaknesses of the project/likely roadblocks", and "Ideas for improvement or specific directions". According to the quality of students' textual feedback, an instructor manually grades the reviews with grade A, B, or C.

| Evaluation Metric | Spearman's $\rho$ | p-value | | Evaluation Metric | Spearman's $\rho$ | p-value |
|---|---|---|---|---|---|---|
| BLEU | 0.023 | 0.772 | | BARTScore-recall | **0.164** | 0.036 |
| ROUGE-L | -0.244 | 0.002 | | LMExaminer | **0.537** | 1.1e-13 |
| BERTScore | -0.061 | 0.439 | | GEM-raw | **0.300** | 9.2e-05 |
| BARTScore-F1 | -0.237 | 0.002 | | GEM | **0.431** | 7.5e-09 |
| BARTScore-prec. | -0.511 | 2.3e-12 | | GEM-S | **0.479** | 7.4e-11 |

Table 1: Spearman's correlation coefficient ($\rho$) and p-values between various evaluation metrics and instructor-annotated grades. Significant positive correlations ($p < 0.05$) are bolded.

**Statistics Metric and Results.** Table 1 shows Spearman's correlation coefficient $\rho$ between the scores of various evaluation metrics and the instructors' grades. All variants of our GEM metrics show a significant positive correlation. The GEM-S has the highest correlation over all three GEM variants, implying the effectiveness of filtering out shortcuts. It also shows competitive performance compared to GPT-4o LMExaminer which has the highest correlation. All other baselines perform worse than GEM, and only BARTScore-Recall has a significant positive correlation. In later sections, we provide further comparisons between GEM and LMExaminer.

## 4.2 RESULT 2: BETTER SENSITIVITY TO DEGRADATION

In this section, we aim to compare how sensitive the evaluation metrics are to degradations that obviously reduce the semantic informativeness of the responses. A statistically significant score decrease indicates the sensitivity of the metric. We first introduce our validation workflow in Algorithm 1, the ICLR peer review dataset, and then introduce several degradation strategies. The workflow and dataset will also be used in the robustness check again manipulation strategies in Section 4.3.

The validation workflow utilizes a dataset $D$ consisting of $n$ tuples $(w_i, x_i, y_i)$ of tasks, candidate responses, and reference responses. For each tuple, we first compute the score $s_i := f(w_i, x_i, y_i)$.

We then replace the response $x_i$ with $x_i'$ according to a degradation/manipulation strategy $M$, and compute $s_i' := f(w_i, x_i', y_i)$. Finally, we compute the means $\mu, \mu'$ and standard deviations $\sigma, \sigma'$ of $\{s_i\}_{i \in [n]}$ and $\{s_i'\}_{i \in [n]}$ respectively. A formal workflow is provided in Algorithm 1 of Appendix D.2.

Note that the evaluation metrics that we compare are in different scales, thus, instead of comparing the mean difference, we standardize the scores with pooled standard deviation $\sigma_{\text{pooled}} := \sqrt{(\sigma^2 + \sigma'^2)/2}$, and compare the standardized mean difference (SMD) $d := \frac{\mu' - \mu}{\sigma_{\text{pooled}}}$ (Andrade, 2020). This form of SMD is also known as Cohen's d (Cohen, 1988).

**ICLR Dataset.** We use the ICLR 2023[3] peer review data publicly available on OpenReview. We randomly select 300 papers as our benchmark dataset, and for each paper, we randomly select 3 original human reviews: one review to serve as a human candidate, and the other two as reference responses.

**Degradation Strategies.** We introduce three degradation strategies as follows. Note that the degraded responses after *Sentence Deletion* or *Deletion & Completion* can be regarded as Blackwell-dominated by the original one. The detailed setup is provided in Appendix D.3.

1. *Sentence Deletion.* We delete every other sentence of $x_i$ to get a degraded response $x_i'$.

2. *Deletion & Completion.* After deletion, we use GPT-4o to complete the deleted sentence with consistent language style and semantics to the original response, which provides an informatively degraded response $x_i'$ but with a similar length to the original one.

3. *Abstract-only Review.* We use Claude-3-sonnet[4] to create a fictitious review $x_i'$ with only the abstract of the paper, and use $x_i'$ to replace $x_i$.

**Statistics and Results.** We compare the standardized mean differences (SMD) of evaluation metrics in Table 2: GEM and GEM-S are only two metrics that consistently have significant negative SMD on all three degradations. The LMExaminer shows the highest effectiveness on Sentence Deletion and Deletion & Completion, but failed to penalize reviews only based on the abstract.

| Evaluation Metric | Sentence Deletion | Deletion & Completion | Abstract-only Review | GPT-4o Rephrase | Llama3.1 Rephrase | Meaningless Elongation |
|---|---|---|---|---|---|---|
| BLEU | **-0.282** | 0.016 | **-0.692** | -0.975 | -0.920 | -0.165 |
| ROUGE-L | **-0.073** | 0.091 | 0.022 | **0.028** | **0.120** | -0.196 |
| BERTScore | **-0.100** | **-0.188** | 0.840 | **0.134** | **0.063** | **0.064** |
| BARTS.-F1 | 0.401 | 0.201 | 0.394 | **0.130** | **0.332** | -0.304 |
| BARTS.-prec. | 0.627 | 0.317 | 0.617 | **0.218** | **0.541** | -0.439 |
| BARTS.-recall | **-0.017** | **-0.016** | 0.004 | -0.027 | -0.028 | **0.004** |
| LMExaminer | **-1.290** | **-0.417** | 0.715 | **0.187** | **0.104** | **0.105** |
| GEM-raw | **-0.123** | **-0.126** | 0.020 | -0.123 | -0.126 | **0.020** |
| GEM | **-0.401** | **-0.308** | **-0.191** | -0.058 | -0.107 | -0.063 |
| GEM-S | **-0.409** | **-0.206** | **-0.566** | -0.046 | -0.114 | -0.070 |

Table 2: Standardized Mean Differences (SMD) $d := (\mu' - \mu)/\sigma_{\text{pooled}}$ of scores after degradations and manipulations of various evaluation metrics. Significant score decreases ($p < 0.05$) after degradations are highlighted in bold green, implying the evaluation metric can **effectively** penalize the degradation. Significant score increases ($p < 0.05$) after manipulations are highlighted in bold red, implying the evaluation metric is **not robust** against the manipulation. We provide 95% confidence interval of SMD values in Appendix A.1.

### 4.3 RESULT 3: BETTER ROBUSTNESS AGAINST MANIPULATION

Evidence has been found that the LLM evaluators, and even humans, are not robust against specific manipulation strategies, they may rely on language style or length as shortcuts instead of providing an evaluation based on semantic quality (Panickssery et al., 2024; Bai et al., 2024; Goldberg et al., 2023), thus, it is important to further check our metric's robustness against several manipulations.

---

[3]At the time of our experiment, ICLR 2024 review data was not yet available through the OpenReview API.

[4]We intentionally use a different LLM from Deletion & Completion to make sure the effect of score decrease is not caused only by the LM.

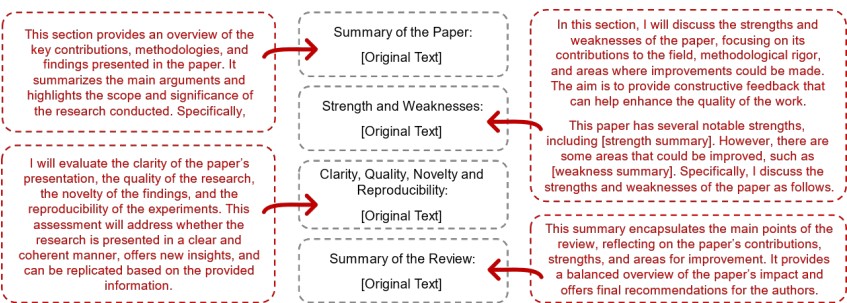

Figure 3: Meaningless Elongation

Specifically, we employ two manipulation strategies, which do not significantly change the semantics, but may change the language style or add meaningless starting sentences. After manipulation, if the score significantly increases, the evaluation metric fails to pass the robustness check. We use the same workflow and dataset introduced in Section 4.2.

**Manipulation Strategies.** We now introduce the manipulation strategies. The detailed setup, including the prompts and examples, is provided in Appendix D.4.

1. *GPT-4o/Llama-3.1 Rephrase.* Following Bai et al. (2024), we use GPT-4o and Llama-3.1 to rephrase the response $x_i$ while trying to keep the semantics unchanged. We test whether the evaluation metrics have preference towards GPT-4o or Llama-3.1 generated language.

2. *Meaningless Elongation.* We adopt a meaningless elongation method similar to Goldberg et al. (2023)'s human subject experiment where a set of fixed meaningless sentences is added to each section of all the reviews as shown in Figure 3.

**Statistics and Results.** The SMDs shown in Table 2 demonstrate that LMexaminer has significant score increases after all three manipulations, indicating its lack of manipulation-resistance. In contrast, GEM and GEM-S exhibit no significant score increase across these manipulations. Furthermore, comparing SMDs after degradations and manipulations in Table 2, we observe that the scores of GEM and GEM-S decrease less after manipulations compared to degradations, indicating that manipulations result in less information loss than degradations.

## 5 GRE-BENCH: BENCHMARKING LLMS WITH GEM

After validating our evaluation metrics GEM and GEM-S in Section 4.1, 4.2, and 4.3, we now show the results of GRE-bench (Generating Review Evaluation Benchmark) based on GEM and GEM-S. We use the same ICLR 2023 peer review dataset introduced in Section 4.2.

We prompt LLMs to generate detailed reviews for each paper in the dataset, with the prompt inspired by Liang et al. (2023), which covers various aspects, including paper summary, strengths, weaknesses, questions, etc. Details of the prompts are provided in Appendix C.3.

We then use GEM and GEM-S to score the LLM-generated reviews. Specifically, as discussed in Section 3.2, we use GEM-S (abstract) and GEM-S (ASSW), where the synopsis consists of an abstract for GEM-S (abstract), and an abstract supplemented with a summary of author-stated strengths and weaknesses (ASSW) for GEM-S (ASSW). We prompt GPT-4o to get the ASSW summary of each paper, the prompt is shown in Appendix C.4. We visualize the results of the GRE-bench based on GEM, GEM-S (abstract), and GEM-S (ASSW) respectively in Figure 4. More detailed numerical results are provided in Table 6 of Appendix A.2.

Figure 4 shows that, within each LLM family, such as the Llama-3 or Claude series, models with larger parameter sizes tend to perform better. Moreover, newer versions tend to outperform their predecessors with similar model sizes, as seen in comparisons between Llama-3.1 and Llama-3. These observations further validate the effectiveness of our GEM and GEM-S metrics in accurately evaluating LLMs.

Notably, several models with large parameter sizes surpass the human baseline (gray line) when using GEM and GEM-S (abstract) metrics. We hypothesize this is because strong LLMs like Claude-3 opus

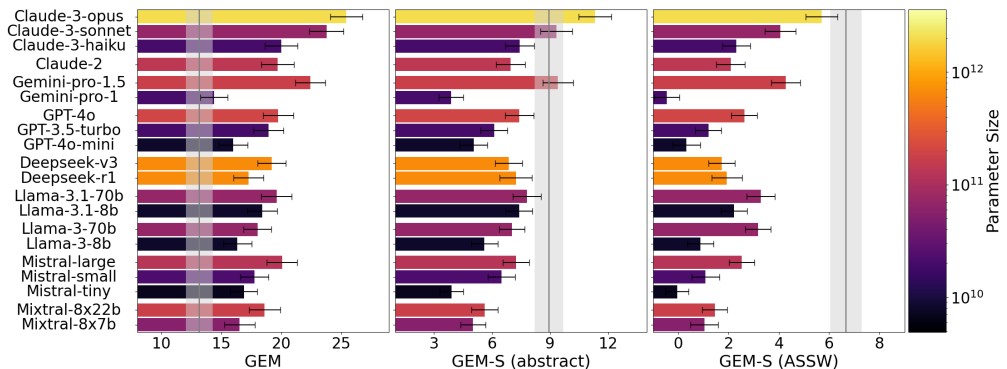

Figure 4: Results of GRE-bench based on three evaluation metrics with 90% confidence intervals vs. model parameter sizes[5] indicated by color. The grey line represents the average human baseline, with the 90% confidence interval shaded in grey.

can effectively retrieve contributions and limitations stated by the authors, and will include that in their reviews. In contrast, human reviewers tend to omit judgments that are less important or already stated by the authors due to writing costs, resulting in a loss of mutual information. This hypothesis is supported by the observation that, when we condition out the author-stated strengths and weaknesses, the human reviewer has the highest GRE-bench score based on GEM-S (ASSW).

This observation further validates the hierarchical information structure discussed in Section 3.2. By conditioning out different levels of information, the benchmark captures subtly different abilities. GEM-S (ASSW) emphasizes the quality of critical judgment, while GEM and GEM-S (abstract) focus more on evaluating LLMs' foundational abilities, including retrieving useful information from the paper, which is especially valuable for differentiating LLMs with smaller parameter size.

In Appendix A.2, we provide more detailed results and discussions of GRE-bench, including GRE-bench's correlations with other benchmarks, its consistency across evaluation LMs (Llama-3.1 8B and Llama-3.1 70B), and detailed numerical results.

# 6 CONCLUSION AND DISCUSSION

In conclusion, we introduce GEM and GEM-S, two metrics for evaluating LLMs on tasks without gold-standard references. They have proved to be accurate and manipulation-resistant, outperforming other metrics in the experiment. Based on GEM and GEM-S, we present GRE-bench for assessing LLMs' peer review capabilities while minimizing the risk of data contamination.

Our approach has certain limitations. First, although our experiments demonstrate its effectiveness, the theoretical results depend on the accurate estimation of the underlying distribution by the LLM, which may not always be reliable. With ongoing advancements in LLM technology, we hope that these estimations will become more accurate over time. Second, while our method does not require gold-standard references, it requires reference that includes original human-generated information. If all reviewers submit solely LLM-generated reviews—such as those only from GPT-4o—the mutual information-based approach may unfairly assign the highest score to GPT-4o. This echoes recent research demonstrating that the absence of original human-generated data in AI evaluation can result in model collapse (Shumailov et al., 2024).

For future work, we can use new open-access peer review data to update GRE-bench every year and observe how the evaluation evolves over time, to monitor any improvements in LLMs' peer review capabilities. This will also ensure the benchmark stays current and minimizes data contamination. Additionally, exploring GEM's potential to evaluate subjective content in other scenarios, such as book reviews and movie reviews, could broaden its applicability and further demonstrate its versatility. We provide initial results in validating GEM on a Yelp restaurant review dataset in Appendix A.4.

---

[5]In Figure 4, the parameter sizes of closed-source models are estimated from unofficial sources. This information is only to provide context, and none of our conclusions or key findings depend on these estimates.

ACKNOWLEDGMENT

We would like to thank Sanzeed Anwar from University of Michigan for data processing. We also thank Zebang Xu and Tianzi Liu from Cornell University for their valuable consultation on the statistical methods employed in our experiments. Additionally, we appreciate the insightful feedback and suggestions provided by the anonymous reviewers at ICLR 2025.

REPRODUCIBILITY STATEMENT

In this section, we demonstrate the efforts we have made to ensure better reproducibility of the paper.

**Complete Theoretical Proof**    In Appendix B, we provide a full proof of the conclusions claimed in the paper.

**Ready-to-Use Code**    We have made all the code for data collection, model fine-tuning, metric computation, and experiments available on GitHub. The repository is at `https://github.com/yx-lu/Benchmarking-LLMs--Judgments-with-No-Gold-Standard`. Additionally, to address potential hardware issues for reproducing the results, our code can be deployed on Google Colab (requiring a T4 or higher GPU, with A100 being recommended). Experiments with Llama-3.1-8B as the evaluation LM are conducted on Google Colab instances with NVIDIA A100 GPU. Experiments in Section 5 employ Llama-3.1-70B as the evaluation LM requiring more GPU VRAM, thus, we use rental instances with a configuration of two NVIDIA H100 GPUs.

**Clear Implementation and Experimental Details**    In Appendix C, we thoroughly discuss the implementation of our GEM and GEM-S metrics. In Appendix D, we provide detailed explanations of the experimental procedures, including how we degrade and manipulate reviews, as well as the methodology for computing the baseline metrics.

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

## A  ADDITIONAL RESULTS

In this section, we provide additional detailed results that were omitted from the main paper and present the results of additional experiments to further validate our GEM metrics.

### A.1  RESULT 2 AND 3 WITH CONFIDENCE INTERVALS

We present the SMDs with 95% confidence intervals of various evaluation metrics under degradations and manipulations in Table 3 and 4.

| Evaluation Metric | Sentence Deletion | Deletion & Completion | Abstract-only Review |
|---|---|---|---|
| BLEU | **-0.282** | 0.016 | **-0.692** |
| | (-0.346,-0.218) | (-0.013,0.046) | (-0.778,-0.607) |
| ROUGE-L | **-0.073** | 0.091 | 0.022 |
| | (-0.122,-0.025) | (0.062,0.121) | (-0.054,0.098) |
| BERTScore | **-0.100** | **-0.188** | 0.840 |
| | (-0.131,-0.069) | (-0.222,-0.155) | (0.769,0.910) |
| BARTScore-F1 | 0.401 | 0.201 | 0.394 |
| | (0.380,0.422) | (0.184,0.218) | (0.344,0.445) |
| BARTScore-precision | 0.627 | 0.317 | 0.617 |
| | (0.595,0.659) | (0.292,0.343) | (0.536,0.698) |
| BARTScore-recall | **-0.017** | **-0.016** | 0.004 |
| | (-0.019,-0.016) | (-0.017,-0.014) | (0.001,0.007) |
| LMExaminer | **-1.290** | **-0.417** | 0.715 |
| | (-1.343,-1.238) | (-0.472,-0.363) | (0.630,0.799) |
| GEM-raw | **-0.123** | **-0.126** | 0.020 |
| | (-0.126,-0.120) | (-0.132,-0.121) | (0.017,0.023) |
| GEM | **-0.401** | **-0.308** | **-0.191** |
| | (-0.448,-0.354) | (-0.358,-0.258) | (-0.261,-0.122) |
| GEM-S | **-0.409** | **-0.206** | **-0.566** |
| | (-0.455,-0.362) | (-0.254,-0.158) | (-0.639,-0.492) |

Table 3: Standardized Mean Differences (SMD) $d := (\mu' - \mu)/\sigma_{\text{pooled}}$ of scores after degradations of various evaluation metrics with 95% confidence intervals in parentheses. Significant score decreases ($p < 0.05$) after degradations are highlighted in bold green, implying the evaluation metric can **effectively** penalize the degradation.

| Evaluation Metric | GPT-4o Rephrase | Llama3.1 Rephrase | Meaningless Elongation |
|---|---|---|---|
| BLEU | -0.975 | -0.920 | -0.165 |
| | (-1.020,-0.930) | (-0.971,-0.870) | (-0.230,-0.101) |
| ROUGE-L | **0.028** | **0.120** | -0.196 |
| | (0.009,0.047) | (0.080,0.160) | (-0.241,-0.151) |
| BERTScore | **0.134** | **0.063** | **0.064** |
| | (0.113,0.155) | (0.032,0.093) | (0.042,0.086) |
| BARTScore-F1 | **0.130** | **0.332** | -0.304 |
| | (0.120,0.140) | (0.299,0.365) | (-0.308,-0.299) |
| BARTScore-precision | **0.218** | **0.541** | -0.439 |
| | (0.204,0.233) | (0.490,0.593) | (-0.446,-0.433) |
| BARTScore-recall | -0.027 | -0.028 | **0.004** |
| | (-0.028,-0.027) | (-0.029,-0.027) | (0.004,0.005) |
| LMexaminer | **0.187** | **0.104** | **0.105** |
| | (0.153,0.221) | (0.060,0.147) | (0.069,0.140) |
| GEM-raw | -0.123 | -0.126 | **0.020** |
| | (-0.126,-0.120) | (-0.132,-0.121) | (0.017,0.023) |
| GEM | -0.058 | -0.107 | -0.063 |
| | (-0.090,-0.026) | (-0.143,-0.070) | (-0.097,-0.030) |
| GEM-S | -0.046 | -0.114 | -0.070 |
| | (-0.079,-0.013) | (-0.149,-0.078) | (-0.104,-0.036) |

Table 4: Standardized Mean Differences (SMD) of scores $d := (\mu' - \mu)/\sigma_{\text{pooled}}$ after manipulations of various evaluation metrics with 95% confidence intervals in parentheses. Significant score increase ($p < 0.05$) after manipulations are highlighted in bold red, implying the evaluation metric are **not robust** against the manipulation.

## A.2 Detailed Results of GRE-bench

**GRE-bench vs. Other benchmarks** The ability to generate informative reviews relies on several key factors, including logical reasoning, critical thinking, and fact-checking. Various widely used benchmarks measure different aspects of LLM performance. To better understand which abilities are most critical for generating high-quality reviews, in the sense of GRE-bench scores, we compared our results with several other benchmarks.

We present the Spearman's correlation coefficients between the GRE-bench scores of all the models and their corresponding scores on other benchmarks[6] in Table 5, providing insights into which factors most strongly influence review generation quality. Notably, GRE-bench highly correlates with HellaSWAG and ARC-C which measures reasoning ability, and less correlates with benchmarks for coding ability (HumanEval) or math ability (MATH, GSM8K).

| Base Metric | MMLU | ARC-C | HellaSwag | GSM8K | MATH | HumanEval | GPQA |
|:---:|:---:|:---:|:---:|:---:|:---:|:---:|:---:|
| GEM | 0.55 | 0.68 | 0.74 | 0.58 | 0.37 | 0.36 | 0.60 |
| GEM-S (abstract) | 0.66 | 0.70 | 0.82 | 0.73 | 0.43 | 0.43 | 0.67 |
| GEM-S (ASSW) | 0.68 | 0.78 | 0.84 | 0.73 | 0.43 | 0.48 | 0.71 |

Table 5: Spearman's correlation coefficients between GRE-bench and other popular benchmarks.

**Consistent results over various Evaluation LMs.** The Spearman's correlation coefficients between the GRE-bench scores based on GEM metric computed using Llama-3.1 8B and that same score computed with Llama-3.1 70B is 0.93. This indicates that the score is largely robust with respect to the specific evaluation LM used. The analogous coefficient for the GRE-bench scores based on GEM-S (abstract) and GEM-S (ASSW) are 0.92 and 0.90 respectively. We further provide the comprehensive results of GRE-bench with Llama-3.1 8B in Table 7. This consistency shows that with our metrics, a small model with 8B parameters that can be deployed on a single NVIDIA L4 GPU, plus proper preprocessing, is still effective in evaluating the quality of judgments generated by much larger models, though larger evaluation LM may have better evaluation performance.

**Numerical results.** We present numerical results of various language models on the GRE-bench based on GEM, GEM-S (abstract), and GEM-S (ASSW) computed with Llama3.1 70B as the evaluation LM, alongside their estimated parameter sizes in Table 6. Note that some estimations of the parameter sizes are from unofficial sources. This information is only to provide context, and none of our conclusions depend on these estimates. The corresponding results of GRE-bench computed with Llama3.1 8B is shown in Table 7.

---

[6]We collect benchmark data from online sources, such as technical reports of language models. Since some models have missing results for specific benchmarks, we exclude those entries when computing correlation. For more details, please refer to our code repository.

| Model Name | Para. Size | GRE-bench | | |
|---|---|---|---|---|
| | (in Billion) | GEM | GEM-S(abs.) | GEM-S(ASSW) |
| deepseek/deepseek-chat | 671 | 19.27 | 6.88 | 1.74 |
| deepseek/deepseek-r1 | 671 | 17.28 | 7.23 | 1.94 |
| openai/gpt-4o-mini (v 20240718) | ∼ 8 | 16.31 | 5.25 | 0.52 |
| openai/gpt-4o (v 20240513) | ∼ 200 | 20.08 | 7.62 | 2.82 |
| openai/gpt-3.5-turbo (v 20240125) | ∼ 20 | 19.28 | 6.31 | 1.40 |
| anthropic/claude-2 | ∼ 137 | 20.04 | 7.17 | 2.27 |
| anthropic/claude-3-haiku (v 20240307) | ∼ 20 | 20.32 | 7.63 | 2.50 |
| anthropic/claude-3-sonnet (v 20240229) | ∼ 70 | 24.11 | 9.53 | 4.25 |
| anthropic/claude-3-opus (v 20240229) | ∼ 2000 | 25.77 | 11.52 | 5.89 |
| google/gemini-pro (v 002) | ∼ 20 | 14.73 | 4.09 | -0.26 |
| google/gemini-pro-1.5 (v 002) | ∼ 175 | 22.76 | 9.61 | 4.47 |
| meta-llama/llama-3-8b-instruct | 8 | 16.68 | 5.81 | 1.07 |
| meta-llama/llama-3-70b-instruct | 70 | 18.35 | 7.23 | 3.37 |
| mistralai/mixtral-8x7b-instruct (v 0.1) | 56 | 16.86 | 5.22 | 1.23 |
| mistralai/mixtral-8x22b-instruct (v 0.1) | 176 | 18.94 | 5.82 | 1.64 |
| meta-llama/llama-3.1-8b-instruct | 8 | 18.74 | 7.60 | 2.40 |
| meta-llama/llama-3.1-70b-instruct | 70 | 19.95 | 8.00 | 3.47 |
| mistralai/mistral-large (v 2407) | 123 | 20.40 | 7.44 | 2.72 |
| mistralai/mistral-small (v 2409) | 22 | 18.08 | 6.70 | 1.28 |
| mistralai/mistral-tiny (v mistral-7b-0.3) | 7 | 17.21 | 4.11 | 0.15 |

Table 6: Results of GRE-bench based on GEM, GEM-S (abstract), and GEM-S (ASSW) with Llama3.1 70B. In the column of parameter size, the symbol ∼ indicates that this size is estimated. For mix-of-expert (MoE) models, we show the total parameter size instead of the activated parameter size.

| Model Name | Para. Size | GRE-bench | | |
|---|---|---|---|---|
| | (in Billion) | GEM | GEM-S(abs.) | GEM-S(ASSW) |
| deepseek/deepseek-chat | 671 | 47.51 | 17.40 | 11.98 |
| deepseek/deepseek-r1 | 671 | 47.45 | 18.89 | 13.80 |
| openai/gpt-4o-mini (v 20240718) | ∼ 8 | 42.55 | 14.44 | 10.25 |
| openai/gpt-4o (v 20240513) | ∼ 200 | 48.01 | 18.73 | 12.78 |
| openai/gpt-3.5-turbo (v 20240125) | ∼ 20 | 45.62 | 15.86 | 10.72 |
| anthropic/claude-2 | ∼ 137 | 55.94 | 24.29 | 17.22 |
| anthropic/claude-3-haiku (v 20240307) | ∼ 20 | 49.77 | 19.58 | 13.34 |
| anthropic/claude-3-sonnet (v 20240229) | ∼ 70 | 56.49 | 24.16 | 16.52 |
| anthropic/claude-3-opus (v 20240229) | ∼ 2000 | 60.94 | 27.94 | 20.21 |
| google/gemini-pro (v 002) | ∼ 20 | 40.65 | 12.86 | 9.49 |
| google/gemini-pro-1.5 (v 002) | ∼ 175 | 52.94 | 22.43 | 16.06 |
| meta-llama/llama-3-8b-instruct | 8 | 43.50 | 15.49 | 11.24 |
| meta-llama/llama-3-70b-instruct | 70 | 46.39 | 18.60 | 13.68 |
| mistralai/mixtral-8x7b-instruct (v 0.1) | 56 | 45.18 | 15.61 | 11.25 |
| mistralai/mixtral-8x22b-instruct (v 0.1) | 176 | 47.29 | 17.03 | 11.85 |
| meta-llama/llama-3.1-8b-instruct | 8 | 46.51 | 18.33 | 12.69 |
| meta-llama/llama-3.1-70b-instruct | 70 | 48.29 | 19.59 | 13.91 |
| mistralai/mistral-large (v 2407) | 123 | 47.84 | 17.84 | 12.52 |
| mistralai/mistral-small (v 2409) | 22 | 45.01 | 15.90 | 11.10 |
| mistralai/mistral-tiny (v mistral-7b-0.3) | 7 | 43.90 | 13.34 | 9.94 |

Table 7: Results of GRE-bench based on GEM, GEM-S (abstract), and GEM-S (ASSW) with Llama3.1 8B. In the column of parameter size, the symbol ∼ indicates that this size is estimated. For mix-of-expert (MoE) models, we show the total parameter size instead of activated parameter size.

## A.3 Consistent Results over Preprocessing Models

To answer the question *"Does using an open-sourced preprocessing model (rather than GPT4o) still lead to effective GEM metric and similar GRE-bench results?"*, we use a Llama3.2-90B-Vision-Instruct model for preprocessing and re-run all validating experiments mentioned in Section 4. We find that most experimental results remain valid except for two manipulation tests. The GRE-bench results also remain highly consistent with the results of using GPT4o preprocessing.

**Positive Correlation with Human Annotation**    GEM still shows a significant positive correlation with human annotation in the proposal peer grading dataset (Section 4.1), however, the Spearman's correlation coefficient between GEM scores and instructor grades drops from 0.433 (GPT4o preprocessing) to 0.274 (Llama3.2-90b preprocessing).

**Sensitivity to Degradation**    In the degradation experiment (Section 4.2), GEM with Llama3.2-90b preprocessing also shows significant sensitivity to all degradations and even performs better in "sentence deletion" than using GPT4o to preprocess.

| Evaluation Metric | Sentence Deletion | Deletion & Completion | Abstract-only Review |
|---|---|---|---|
| GEM (Llama Preproccessing) | **-0.566** (-0.625,-0.507) | **-0.317** (-0.371,-0.263) | **-0.078** (-0.154,-0.002) |
| GEMS (Llama Preproccessing) | **-0.549** (-0.607,-0.491) | **-0.205** (-0.258,-0.153) | **-0.327** (-0.399,-0.254) |

Table 8: Standardized Mean Differences (SMD) $d := (\mu' - \mu)/\sigma_{\text{pooled}}$ of scores after degradations of GEM and GEM-S metrics (preprocessing with Llama-3.2 90B) with 95% confidence intervals in parentheses. Significant score decreases ($p < 0.05$) after degradations are highlighted in bold green, implying the evaluation metric can **effectively** penalize the degradation.

**Weaker Robustness than GPT-4o Preprocessing**    In the manipulation experiments (Section 4.2), GEM with Llama3.2-90b preprocessing is robust against "GPT4 rephrase" but fails in the "Llama-3.1 rephrase" and "Meaningless Elongation".

| Evaluation Metric | GPT-4o Rephrase | Llama3.1 Rephrase | Meaningless Elongation |
|---|---|---|---|
| GEM (Llama Preproccessing) | -0.132 (-0.180,-0.084) | **0.124** (0.075,0.173) | **0.138** (0.086,0.189) |
| GEM-S (Llama Preproccessing) | -0.126 (-0.173,-0.079) | **0.156** (0.109,0.203) | **0.112** (0.061,0.163) |

Table 9: Standardized Mean Differences (SMD) $d := (\mu' - \mu)/\sigma_{\text{pooled}}$ of scores after manipulations of GEM and GEM-S metrics (preprocessing with Llama-3.2 90B) with 95% confidence intervals in parentheses. Significant score increases ($p < 0.05$) after manipulations are highlighted in bold red, implying the evaluation metrics are **not robust** against the manipulation.

**Consistent GRE-bench Results**    Based on the GEM metric with llama3.2-90b preprocessing, we re-compute the GRE-bench scores with ICLR 2023 dataset (Section 5). The Spearman's correlation coefficient between the GRE-bench scores based on the GEM metric with Llama3.2-90B preprocessing and the scores with GPT-4o preprocessing shown in Table 7 is 0.89. The analogous coefficient for the GRE-bench scores based on GEM-S (abstract) and GEM-S (ASSW) are 0.85 and 0.83 respectively. All detailed numerical results are provided in Table 10.

| Model Name | Para. Size | GRE-bench | | |
|---|---|---|---|---|
| | (in Billion) | GEM | GEM-S(abs.) | GEM-S(ASSW) |
| openai/gpt-4o-mini (v 20240718) | $\sim 8$ | 45.76 | 19.58 | 18.27 |
| openai/gpt-4o (v 20240513) | $\sim 200$ | 50.17 | 23.12 | 20.55 |
| openai/gpt-3.5-turbo (v 20240125) | $\sim 20$ | 48.56 | 20.92 | 19.04 |
| anthropic/claude-2 | $\sim 137$ | 54.61 | 26.07 | 23.93 |
| anthropic/claude-3-haiku (v 20240307) | $\sim 20$ | 52.70 | 24.87 | 22.18 |
| anthropic/claude-3-sonnet (v 20240229) | $\sim 70$ | 54.83 | 25.82 | 22.78 |
| anthropic/claude-3-opus (v 20240229) | $\sim 2000$ | 62.08 | 33.13 | 30.53 |
| google/gemini-pro (v 002) | $\sim 20$ | 46.75 | 21.33 | 20.43 |
| google/gemini-pro-1.5 (v 002) | $\sim 175$ | 56.36 | 27.40 | 24.44 |
| meta-llama/llama-3-8b-instruct | 8 | 47.36 | 20.96 | 19.86 |
| meta-llama/llama-3-70b-instruct | 70 | 55.62 | 29.61 | 27.98 |
| mistralai/mixtral-8x7b-instruct (v 0.1) | 56 | 50.53 | 22.83 | 21.08 |
| mistralai/mixtral-8x22b-instruct (v 0.1) | 176 | 51.57 | 23.84 | 21.79 |
| meta-llama/llama-3.1-8b-instruct | 8 | 51.04 | 24.51 | 22.28 |
| meta-llama/llama-3.1-70b-instruct | 70 | 50.26 | 23.68 | 21.73 |
| mistralai/mistral-large (v 2407) | 123 | 51.81 | 23.59 | 21.20 |
| mistralai/mistral-small (v 2409) | 22 | 50.22 | 22.99 | 20.58 |
| mistralai/mistral-tiny (v mistral-7b-0.3) | 7 | 48.42 | 20.04 | 18.86 |

Table 10: Results of GRE-bench based on GEM, GEM-S (abstract), and GEM-S (ASSW) with Llama3.2 90B preprocessing. In the column of parameter size, the symbol $\sim$ indicates that this size is estimated.

Overall, as more advanced models can preprocess reports more accurately, the GEM metric will be more accurate and manipulation-resistant with superior preprocessing. Considering the frequent release and update of LLMs, utilizing more advanced LLMs for preprocessing enhances both the accuracy and robustness of the GEM metric. Consequently, our GEM metrics stand to benefit from the continuous advancements in LLM technology.

## A.4 VALIDATION ON YELP RESTAURANT REVIEW DATASET

To further validate our GEM metrics' effectiveness in evaluating judgments without gold-standard references in scenarios in addition to peer review. We use an open-access online business review data from Yelp, and repeat the same workflow in Section 4.2 and 4.3, to check how sensitive GEM metrics are to degradations and how robust GEM metrics are to manipulations.

The results are similar to those observed in experiments on ICLR dataset (Section 4.2 and 4.3). Our GEM and GEM-S demonstrates significant sensitivity to degradations and robustness against manipulations.

**Yelp Review**   We randomly sample 300 restaurants from the entire dataset which have at least 50 reviews with at least 1000 characters. For each restaurant, we randomly sample 3 customer reviews among all reviews with at least 1000 characters. Yelp reviews quality varies much more than ICLR reviews, thus, we use length as a heuristic filter to get reasonable quality reviews.

**GEM Setup**   We use GPT-4o for text preprocessing, and use Llama-3.1 8B as the evaluation-LM. For GEM-S, we use the categories of the restaurant as the synopsis, which is given by the Yelp dataset.

**GEM demonstrates sensitivity to degradations.**   We reuse the *Sentence Deletion* and *Deletion & Completion* degradation methods described in Section 4.2. However, since restaurant reviews lack "abstract", the *Abstract-only Review* degradation cannot be applied. Instead, we propose *Random Replacement* degradation, where we degrade the original review $x_i$ by replacing it with a random review $x'_i$ of another randomly selected restaurant.

Table 11 presents the standardized mean differences (SMD) with 95% confidence intervals for GEM and baseline metrics before and after these degradations. The results align with those observed in the ICLR dataset (Section 4.2): GEM and GEM-S consistently exhibit significant SMD across all three degradations. Additionally, while LMExaminer is more sensitive to degradations like *Sentence Deletion* and *Deletion & Completion*, which impair presentation quality, GEM shows greater sensitivity to the *Random Replacement* degradation, which severely compromises semantic informativeness.

| Evaluation Metric | Sentence Deletion | Deletion & Completion | Random Replacement |
|---|---|---|---|
| BLEU | **-0.348** | **-0.113** | **-0.231** |
|  | (-0.436,-0.260) | (-0.200,-0.026) | (-0.355,-0.107) |
| ROUGE-L | **-0.677** | 0.009 | **-0.345** |
|  | (-0.765,-0.589) | (-0.064,0.081) | (-0.462,-0.229) |
| BERTscore | **-0.318** | 0.011 | **-0.722** |
|  | (-0.359,-0.276) | (-0.032,0.054) | (-0.836,-0.609) |
| BARTscore-F1 | 0.744 | 0.457 | -0.042 |
|  | (0.705,0.784) | (0.427,0.487) | (-0.163,0.079) |
| BARTscore-precision | 0.877 | 0.536 | -0.023 |
|  | (0.831,0.924) | (0.501,0.571) | (-0.159,0.113) |
| BARTscore-recall | **-0.015** | **-0.014** | **-0.046** |
|  | (-0.018,-0.012) | (-0.017,-0.011) | (-0.051,-0.041) |
| LMExaminer | **-1.245** | **-0.466** | **-0.204** |
|  | (-1.328,-1.162) | (-0.550,-0.382) | (-0.340,-0.068) |
| GEM | **-0.386** | **-0.238** | **-0.516** |
|  | (-0.459,-0.314) | (-0.318,-0.159) | (-0.624,-0.408) |
| GEM-S | **-0.361** | **-0.195** | **-0.283** |
|  | (-0.436,-0.287) | (-0.277,-0.114) | (-0.385,-0.182) |

Table 11: Standardized Mean Differences (SMD) $d := (\mu' - \mu)/\sigma_{\text{pooled}}$ of scores after degradations of various evaluation metrics with 95% confidence intervals in parentheses. Significant score decreases ($p < 0.05$) after degradations are highlighted in bold green, implying the evaluation metric can **effectively** penalize the degradation.

**GEM demonstrates robustness to manipulation.**   Similarly, we reuse the *GPT-4o Rephrase* and *Llama-3.1 Rephrase* manipulations to test whether our metric is robust against specific language styles. However, since Yelp restaurant reviews lack a structured format like ICLR peer reviews (as illustrated in Figure 3), it is unclear how to construct a suitable *Meaningless Elongation* manipulation.

Table 12 shows the standardized mean differences (SMD) with 95% confidence intervals for GEM and baseline metrics before and after these manipulations. The results remain similar to those observed in Section 4.3. GEM metrics remain robust against both types of manipulation, while LMExaminer shows a significant increase in scores after LLM rephrasing.

| Evaluation Metric | GPT-4o Rephrase | Llama3.1 Rephrase |
|---|---|---|
| BLEU | **0.100** | 0.046 |
| | (0.014,0.185) | (-0.063,0.156) |
| ROUGE-L | 0.020 | **0.083** |
| | (-0.037,0.076) | (0.008,0.157) |
| BERTscore | -0.016 | -0.098 |
| | (-0.041,0.008) | (-0.133,-0.064) |
| BARTscore-F1 | **0.316** | **0.425** |
| | (0.287,0.346) | (0.371,0.478) |
| BARTscore-precision | **0.371** | **0.508** |
| | (0.337,0.405) | (0.445,0.571) |
| BARTscore-recall | -0.012 | -0.016 |
| | (-0.013,-0.010) | (-0.018,-0.014) |
| LMExaminer | **0.300** | **0.453** |
| | (0.240,0.361) | (0.387,0.518) |
| GEM | 0.021 | 0.010 |
| | (-0.035,0.076) | (-0.055,0.075) |
| GEM-S | 0.020 | -0.012 |
| | (-0.037,0.077) | (-0.080,0.055) |

Table 12: Standardized Mean Differences (SMD) of scores $d := (\mu' - \mu)/\sigma_{\text{pooled}}$ after manipulations of various evaluation metrics with 95% confidence intervals in parentheses. Significant score increases ($p < 0.05$) after manipulations are highlighted in bold red, implying the evaluation metrics are **not robust** against the manipulation.

### A.5 EXPLORATION OF PROPERTIES OF LLM ESTIMATED DISTRIBUTION

An interesting question raised by our anonymous reviewer is:

*As we know* $\Pr[x,y] = \Pr[x] \cdot \Pr[y|x] = \Pr[y] \cdot \Pr[x|y]$, *does this symmetry also applies to the LLM estimated probability, i.e.,* $\Pr_{\text{LLM}}[x] \cdot \Pr_{\text{LLM}}[y|x] = \Pr_{\text{LLM}}[y] \cdot \Pr_{\text{LLM}}[x|y]$?

To answer this question, we randomly select 500 papers from ICLR 2023 and 1874 corresponding reviews in total. For each pair of reviews $x$, $y$ for the same paper, we use our method to compute $\log \Pr_{\text{LLM}}[x|y]$, $\log \Pr_{\text{LLM}}[y|x]$, $\log \Pr_{\text{LLM}}[x]$, and $\log \Pr_{\text{LLM}}[y]$. For those 5328 pairs, we compute the Spearman correlation coefficient between $\Pr_{\text{LLM}}[x] \cdot \Pr_{\text{LLM}}[x|y] := \exp(\log\Pr_{\text{LLM}}[x|y] + \log\Pr_{\text{LLM}}[y])$ and $\Pr_{\text{LLM}}[y] \cdot \Pr_{\text{LLM}}[y|x] := \exp(\log\Pr_{\text{LLM}}[y|x] + \log\Pr_{\text{LLM}}[x])$. The Spearman correlation coefficient is 0.943, which indicates that they are highly positively correlated, despite minor differences. Potential noise in the estimation process may account for these differences.

The results revealed a high positive correlation between these two values, although they were not precisely equal. Despite this difference, the empirical results in our paper indicate that this does not negatively impact the desired properties, including the accuracy and manipulation resistance.

## B   OMITTED PROOFS

**Proposition 3.1.** *When the KL-divergence[7] between the LLM estimated distribution and the underlying distribution satisfies*

$$D_{KL}\Big[\Pr_{\mathrm{LLM}}[Y=\cdot\,|\,X=x_H]\Big\|\Pr[Y=\cdot\,|\,X_H=x_H]\Big]<\epsilon,\,\forall x_H.$$

*For the two candidates H and L discussed above, the information structure of H Blackwell dominates L's, when the size of dataset $n$ goes to infinity, we have $\widehat{I}(X_H;Y)\geq\widehat{I}(X_L;Y)-\epsilon$.*

*Proof of Proposition 3.1.* By the definition of estimated PMI, $\widehat{\mathrm{PMI}}(x;y)=\log\Pr_{\mathrm{LLM}}[Y=y\,|\,X=x]-\log\Pr_{\mathrm{LLM}}[Y=y]$, we have

$$\widehat{I}(X;Y)=\frac{1}{n}\sum_{i=1}^{n}\widehat{\mathrm{PMI}}(x_i;y_i)$$

$$=\frac{1}{n}\sum_{i=1}^{n}\log\Pr_{\mathrm{LLM}}[Y=y_i\,|\,X=x_i]-\log\Pr_{\mathrm{LLM}}[Y=y_i]$$

$$=\frac{1}{n}\sum_{i=1}^{n}\log\Pr_{\mathrm{LLM}}[Y=y_i\,|\,X=x_i]-\frac{1}{n}\sum_{i=1}^{n}\log\Pr_{\mathrm{LLM}}[Y=y_i].$$

Note that $\frac{1}{n}\sum_{i=1}^{n}\log\Pr_{\mathrm{LLM}}[Y=y_i]$ does not change whatever the candidate response follows $X_H$ or $X_L$. In addition, recall that we assume that all task-response tuples $(w_i,x_i,y_i)$ are i.i.d. sampled from the underlying distribution $\mathcal{D}$. Therefore, with the law of large numbers, we have $\widehat{I}(X_H;Y)\geq\widehat{I}(X_L;Y)-\epsilon$ further equivalent to

$$\mathbb{E}\Big[\log\Pr_{\mathrm{LLM}}[Y=y\,|\,X=x_H]\Big]\geq\mathbb{E}\Big[\log\Pr_{\mathrm{LLM}}[Y=y\,|\,X=x_L]\Big]-\epsilon.$$

On the left hand side, we have

$$\mathbb{E}\Big[\log\Pr_{\mathrm{LLM}}[Y=y\,|\,X=x_H]\Big]$$

$$=\sum_{x_H}\Pr[X_H=x_H]\sum_{y}\Pr[Y=y\,|\,X_H=x_H]\log\Pr_{\mathrm{LLM}}[Y=y\,|\,X=x_H]$$

$$\geq\sum_{x_H}\Pr[X_H=x_H]\left(\sum_{y}\Pr[Y=y\,|\,X_H=x_H]\log\Pr[Y=y\,|\,X_H=x_H]-\epsilon\right)$$

(This step follows the condition of bounded KL-divergence in Proposition 3.1.)

$$=\sum_{x_H,y}\Pr[X_H=x_H,Y=y]\log\Pr[Y=y\,|\,X_H=x_H]-\epsilon$$

$$=-H(Y\,|\,X_H)-\epsilon$$

$$=I(Y;X_H)-H(Y)-\epsilon.$$

On the right hand side, we have

$$\mathbb{E}\Big[\log\Pr_{\mathrm{LLM}}[Y=y\,|\,X=x_L]\Big]$$

$$=\sum_{x_L}\Pr[X_L=x_L]\sum_{y}\Pr[Y=y\,|\,X_L=x_L]\log\Pr_{\mathrm{LLM}}[Y=y\,|\,X=x_L]$$

$$\leq\sum_{x_L}\Pr[X_L=x_L]\sum_{y}\Pr[Y=y\,|\,X_L=x_L]\log\Pr[Y=y\,|\,X_L=x_L]$$

(This step follows the fact that logarithm is a proper scoring rule (Hendrickson & Buehler, 1971).)

$$=-H(Y\,|\,X_L)$$

$$=I(Y;X_L)-H(Y).$$

---

[7]The KL-divergence between two distributions over the same probability space is $D_{\mathrm{KL}}(P\|Q)=\sum_{x}P(x)\log(P(x)/Q(x))$.

Since $X_H$ follows an information structure $\sigma_H$ Blackwell dominates $\sigma_L$ that $X_L$ follows, by the information process inequality, we have $I(Y;X_H) > I(Y;X_L)$.

Therefore, combining all these together, we have

$$\mathbb{E}\left[\log \Pr_{LLM}[Y=y \,|\, X=x_H]\right] \geq \mathbb{E}\left[\log \Pr_{LLM}[Y=y \,|\, X=x_L]\right] - \epsilon$$

and consequently, we have $\widehat{I}(X_H;Y) \geq \widehat{I}(X_L;Y) - \epsilon$.

$\square$

**Proposition B.1.** *For a given random variable $Z$, when the KL-divergence between the LLM estimated distribution and the underlying distribution satisfies*

$$D_{KL}\left[\Pr_{LLM}[Y=\cdot \,|\, X=x_H, Z=z] \,\Big\|\, \Pr[Y=\cdot \,|\, X_H=x_H, Z=z]\right] < \epsilon, \,\forall x_H$$

*For the two candidates $H$ and $L$ discussed above, and the information structure of $H$ Blackwell dominates $L$'s, when the size of dataset $n$ goes to infinity, we have*

$$\widehat{I}(X_H;Y \,|\, Z) \geq \widehat{I}(X_L;Y \,|\, Z) - \epsilon$$

*Proof.* By following similar steps of the proof of Proposition 3.1, we reduce the problem to showing

$$\mathbb{E}\left[\log \Pr_{LLM}[Y=y \,|\, X=x_H, Z=z]\right] \geq \mathbb{E}\left[\log \Pr_{LLM}[Y=y \,|\, X=x_L, Z=z]\right] - \epsilon.$$

Again, following similar steps, we have that the left hand side follows

$$\mathbb{E}\left[\log \Pr_{LLM}[Y=y \,|\, X=x_H, Z=z]\right]$$

$$\geq \sum_z \Pr[Z=z](I(Y;X_H \,|\, Z=z) - H(Y \,|\, Z=z) - \epsilon)$$

$$= I(Y;X_H \,|\, Z) - H(Y \,|\, Z) - \epsilon$$

Similarly, we have the right hand side

$$\mathbb{E}\left[\log \Pr_{LLM}[Y=y \,|\, X=x_L, Z=z]\right]$$

$$\leq \sum_z \Pr[Z=z](I(Y;X_L \,|\, Z=z) - H(Y \,|\, Z=z) - \epsilon)$$

$$= I(Y;X_L \,|\, Z) - H(Y \,|\, Z)$$

Therefore, combining all these together, we have

$$\widehat{I}(X_H;Y \,|\, Z) \geq \widehat{I}(X_L;Y \,|\, Z) - \epsilon$$

$\square$

## C  IMPLEMENTATION DETAILS

In this section, we provide a detailed interpretation of how we implement our proposed metrics to facilitate the replication of our results. This includes comprehensive descriptions of all prompts utilized for generating peer-review judgments, preprocessing the judgments, predicting the judgments, as well as other details that help reproduce the implementation. In all LLM API calls in our experiments, for reproducibility, we keep the temperature at 0 and the maximum (output) token count at 4000, unless otherwise specified.

### C.1  JUDGMENTS PREPROCESSING

To filter out the shortcuts in the review judgments, we use a preprocessing procedure. We employ an LLM to rewrite the original review judgments in a certain compressed format, eliminating the impact of various aspects such as semantics, syntax, and language styles. Here, we employ the LLM GPT-4o (specifically, gpt-4o-2024-05-13) to preprocess the judgments generated by all candidate LLMs (and human reviewers). Here is the prompt we use:

---

**System Prompt**

Carefully read the text of a scientific paper review. You should summarize each evaluation in the review in a separate line. Begin each summary line with one of the following phrases: 'The reviewer appreciates', 'The reviewer criticizes', 'The reviewer questions', 'The reviewer suggests'. You need to keep the summary as concise as possible, excluding specific details about the paper's content, such as topics, ideas, methods, findings, and any mathematical symbols.

You should ensure that even if multiple evaluations are mentioned in the same sentence in the original review, you should still split it into separate lines. For example, you should not output a line like 'The reviewer appreciates the well-written paper and good experimental performance'. In contrast, you should output 'The reviewer appreciates the well-written paper' and 'The reviewer appreciates good experimental performance' in two lines.

**User Prompt**

{Original Review Judgments}

---

### C.2  JUDGMENTS PREDICTION

We compute our metric, the Generative Estimator for Mutual Information (GEM), by utilizing both the original and fine-tuned versions of Llama-3.1-8B-Instruct[8] to predict the conditional probability $\Pr_{\text{LLM}}[Y = y \mid X = x]$ and the marginal probability $\Pr_{\text{LLM}}[Y = y]$. In this context, $x$ and $y$ represent preprocessed review judgments. The same methodology is employed for calculating the GEM-S score. Below is the prompt used for prediction.

---

**System Prompt**

You are the second reviewer for a scientific paper. You are given the abstract of the paper and a list of review judgments from the first reviewer, starting with 'The reviewer appreciates/criticizes/questions/suggests'. Your task is to provide your own judgments of the paper based on the given materials. You should create a separate line for each judgment you have, starting with 'The reviewer appreciates/criticizes/questions/suggests'. Ensure your judgments are concise, excluding specific details about the paper's content.

**User Prompt**

[Abstract of the paper]

{Abstract of the Paper if the metric is GEM-S, and "Not Available" if the metric is GEM}

[Review judgments from the first reviewer]

{Preprocessed Review Judgments ($x$) for conditional probability, and "Not Available" for marginal probability}

**Forced LLM Output**

{Preprocessed Review Judgments ($y$)}

---

[8]We employ 4-bit quantization.

## C.3 Peer-review Judgments Generation in GRE-bench

We provide the implementation details of GRE-bench where we employ academic peer review as the task to evaluate LLMs' ability to generate informative judgments. Following Liang et al. (2023)'s research about LLM-generated peer review, we use the Sciencebeam pdf parser to convert the ICLR submissions in pdf format to text format, and removed the references and all appendices. When using the sciencebeam PDF parser, it retains all image and table titles and captions while disregarding other content. While a few ICLR submissions require OCR before parsing into text (5 out of all submissions in 2023), none of the 300 papers we experimented with required OCR, ensuring no OCR errors. Additionally, we have verified that the parsed text of papers in our dataset does not exceed the input token limit of the LLM.

While we acknowledge that PDF parsing is an inherently imperfect process that may occasionally introduce artifacts or miss certain elements, our experimental design mitigates potential biases because all LLM models in our evaluation receive identical parsed versions of each paper, and any parsing inconsistencies would affect all models equally, making the relative performance comparisons robust to such noise.

Below is the prompt utilized to request each candidate LLM to produce its review judgments.

---

**System Prompt**

You are given a paper submission for a top-tier Machine Learning conference which you need to write a detailed review. Please read the paper carefully. Once you have finished reading, please provide the following in your review:

First, write a concise summary of the key points and contributions of the paper inside <summary> tags.

Next, think critically about the strengths and weaknesses of the submission. Inside <strengths> tags, give a numbered list of at least 4 key reasons why this paper should potentially be accepted to the conference. For each reason, use sub-bullet points to provide detailed arguments and evidence from the paper to support that reason.

Then, inside <weaknesses> tags, give a numbered list of at least 4 key reasons why this paper should potentially be rejected from the conference. Again, for each reason, use sub-bullet points to provide detailed arguments and evidence from the paper to support that reason.

After weighing the reasons for and against, think of some open questions you have about the work. List your questions inside <questions> tags.

Remember, as a reviewer your job is to rigorously evaluate the strengths and weaknesses of the work and to provide critical but constructive feedback to the authors. Be thorough, specific and detailed in your arguments and feedback. Highlight both the positives and negatives you see, and justify your points carefully with reference to the content of the paper.

**User Prompt**

```
{Full paper in Text Format}
```

---

## C.4 Summary of Author-stated Strengths and Weaknesses

In GEM-S (ASSW), we need to embed the strength and weakness text stated by the authors into the input of LLM. To do this, we employ gpt-4o to summarize it using the following prompt.

**System Prompt**

You are given a paper submission for a top-tier Machine Learning conference. Your goal is to identify and list the strengths and weaknesses that the paper claims about itself. This task requires careful reading of the paper.

Please follow these steps to complete the task:

1. Carefully read the entire paper submission. As you read, identify instances where the authors mention strengths or positive aspects of their research, methodology, results, or contributions. These are the strengths claimed by the paper. Also, identify instances where the authors mention limitations, weaknesses, or areas for future improvement in their work. These are the weaknesses claimed by the paper.

2. Compile your findings into two separate lists: one for strengths and one for weaknesses.

3. For each list, write each point on a separate line, keeping it concise. Add an extra blank line between each point for clarity.

4. Format your output as follows:

¡strengths_claimed_by_the_paper¿

[List each strength claimed by the paper in separate lines, with an extra blank line between each point]

¡/strengths_claimed_by_the_paper¿

¡weaknesses_claimed_by_the_paper¿

[List each weakness claimed by the paper in separate lines, with an extra blank line between each point]

¡/weaknesses_claimed_by_the_paper¿

Important: Focus only on the strengths and weaknesses that the paper claims about itself. Do not include your own evaluation or opinion of the paper's merits or shortcomings. Do not include the strengths and weaknesses of the baseline. Your task is to report what the authors themselves have stated about their work's strengths and limitations.

**User Prompt**

{Full paper in Text Format}

# D    EXPERIMENTAL DETAILS

In this section, we provide detailed information regarding the experiments we conducted. This includes the specific processes used to degrade and manipulate reviews, as well as the implementation of the baseline metrics.

## D.1    BASELINE METRICS

In the paper, we use BLEU, ROUGE-L, BERTScore and LMExaminer as the baseline metrics. BLEU and ROUGE-L are both detail free. We employ NLTK as the tokenizer and utilized existing library functions to compute the BLEU and ROUGE-L scores.

*BERTScore* (Zhang et al., 2019) evaluates generated text by measuring the cosine similarity between token embeddings from a pre-trained language model. Instead of BERT, we use a recent model stella_en_400M_v5 (Zhang, 2024) as the evaluation-LM for better token embedding performance.

*BARTScore* (Yuan et al., 2021) is a probability-based metric. It has three variants, precision, recall, and F1-score. Instead of BART, we use the state-of-the-art Llama3.1-8b as the evaluation-LM for estimating the probability, which is the same as our GEM metrics.

*LMExaminer* (Bai et al., 2024) is one of the recently popular methods where a language model (LM) is used to evaluate the quality of generated text based on specific evaluation criteria, serving as an examiner. With empirical evidence suggesting that GPT-4 outperforms open-source models and even finetuned models of pointwise grading on AlignBench (Ke et al., 2023), we use GPT-4o as a baseline in our experiment. For peer review tasks, our prompt adopts criteria based on Review Quality Indicators (RQIs) (Goldberg et al., 2023; Van Rooyen et al., 1999; Superchi et al., 2019) including four aspects, understanding, coverage, substantiation, constructiveness.

We input the full text of the paper along with the review to be judged into the LLM and prompt it to score based on RQIs. We show our prompts below.

**System Prompt**

You are an expert tasked with evaluating the quality of a review for a Machine Learning paper. Your goal is to assess how well the review critiques the paper and provides valuable feedback to the authors. The paper will be given after '[Paper]' and the review will be given after '[Review]'.

To judge the quality of this review, consider the following criteria:

1. Understanding: Does the reviewer demonstrate a clear understanding of the paper's main contributions, methodology, and results?

2. Coverage: Does the review address all major aspects of the paper, including the problem statement, methodology, experiments, results, and conclusions?

3. Substantiation: Does the reviewer provide specific examples or references from the paper to support their comments and criticisms?

4. Constructiveness: Does the review offer helpful suggestions for improvement or identify areas where the paper could be strengthened?

For each criterion, provide a detailed analysis of how well the review meets the standard.

After analyzing each criterion, provide an overall assessment of the review's quality. Consider how well it serves its purpose of offering valuable feedback to the authors.

Finally, assign a score to the review on a scale of 0 to 10, where 0 is the lowest quality, 5 is the average quality, and 10 is the highest quality.

Present your evaluation in the following format:

<analysis>

[Your detailed analysis here]

</analysis>

<overall_assessment>

[Your overall assessment here]

</overall_assessment>

<overall_score>

[Your final quality score here]

</overall_score>

**User Prompt**

[Paper]

`{Full paper in Text Format}`

[Review]

`{Original Review Judgments}`

## D.2  VALIDATION WORKFLOW FOR DEGRADATIONS AND MANIPULATIONS

---
**ALGORITHM 1:** Validation Workflow

---
**Input:** A dataset $D$ with $n$ tuples of tasks and associated text responses.
   An evaluation metric $f$ computing the scores. A degradation/manipulation strategy $M$.
**Output:** Statistics metrics.

**for** $i=1$ **to** $n$ **do**
   Get the $i$-th tuple from the dataset: task $w_i$, candidate response $x_i$, and reference response $y_i$ ;
   Compute $s_i := f(w_i, x_i, y_i)$;

   Replace the response $x_i$ with $x_i'$ according to degradation/manipulation strategy $M$;
   Compute $s_i' := f(w_i, x_i', y_i)$;
**end**
Compute the means $\mu, \mu'$ and standard deviations $\sigma, \sigma'$ of $\{s_i\}_{i \in [n]}$ and $\{s_i'\}_{i \in [n]}$ respectively.

---

## D.3 DEGRADATIONS

### D.3.1 REVIEW DEGRADATION: SENTENCE DELETION

The ICLR 2023 review comments must follow a specific formatting with four sections including "Summary Of The Paper", "Strengths And Weaknesses", "Clarity, Quality, Novelty And Reproducibility" and "Summary Of The Review". When performing sentence deletion, we maintain the original format unchanged. Other sentences are identified based on periods, question marks, exclamation points, and line breaks, which mark the end of a sentence. In each section, we retain all sentences with odd-numbered positions and delete those in even-numbered positions. Below is a toy example for illustration.

---

**Before Deletion**

Summary Of The Paper:

This is the first sentence. This is the second sentence. This is the third sentence.

Strengths And Weaknesses:

This is the first sentence. This is the second sentence. This is the third sentence. This is the fourth sentence.

Clarity, Quality, Novelty And Reproducibility:

This is the first sentence.

Summary Of The Review:

This is the first sentence. This is the second sentence.

**After Deletion**

Summary Of The Paper:

This is the first sentence. This is the third sentence.

Strengths And Weaknesses:

This is the first sentence. This is the third sentence.

Clarity, Quality, Novelty And Reproducibility:

This is the first sentence.

Summary Of The Review:

This is the first sentence.

---

### D.3.2 REVIEW DEGRADATION: DELETION & COMPLETION

When applying the deletion & completion method to degrade reviews, we initially delete half of the sentences, similar to the sentence deletion process, and replace the omitted content with "[There is one missing sentence]". Subsequently, we utilize GPT-4o to complete these sentences. Below is a toy example.

---

**Before Deletion**

Summary Of The Paper:

This is the first sentence. This is the second sentence. This is the third sentence.

Strengths And Weaknesses:

This is the first sentence. This is the second sentence. This is the third sentence. This is the fourth sentence.

Clarity, Quality, Novelty And Reproducibility:

This is the first sentence.

Summary Of The Review:

This is the first sentence. This is the second sentence.

**After Deletion**

Summary Of The Paper:

This is the first sentence. [There is one missing sentence] This is the third sentence

Strengths And Weaknesses:

This is the first sentence. [There is one missing sentence] This is the third sentence. [There is one missing sentence]

Clarity, Quality, Novelty And Reproducibility:

This is the first sentence.

Summary Of The Review:

This is the first sentence. [There is one missing sentence]

---

Here, we present the prompt we utilize to leverage GPT-4o for completing the deleted sentences.

**System Prompt**

You are tasked with completing missing sentences of a peer review evaluation given by the user while keeping all its existing sentences. Your goal is to complete all missing sentences indicated by [There is one missing sentence] of this peer review evaluation, following these rules:

- Insert new sentences that don't contribute significant information.

- Place these sentences between existing sentences where they seem most natural.

- Keep the overall language style similar.

- Ensure these added sentences don't contradict or substantially alter the original content.

Remember to maintain the essence and order of the original content while completing all missing sentences indicated by [There is one missing sentence].

**User Prompt**

{Judgments after Deletion}

### D.3.3 REVIEW DEGRADATION: ABSTRACT-ONLY REVIEW

In this degradation process, we utilize claude-3-sonnet to generate a review based solely on the paper's abstract, which serves as a substitute for the original review. Below is the prompt we use to complete this task.

**System Prompt**

You are given an abstract of a paper submission for a top-tier Machine Learning conference. You need to write a detailed peer review of the paper with only the abstract. Please read the abstract carefully. Once you have finished reading, please provide the following in your review:

First, write a concise summary of the key points and contributions of the paper in "Summary Of The Paper" section.

Next, in the "Strength And Weaknesses" section, think critically about the strengths and weaknesses of the submission. Give a numbered list of at least 4 key reasons why this paper should potentially be accepted to the conference. For each reason, use sub-bullet points to provide detailed arguments and evidence from the paper to support that reason. Then, give a numbered list of at least 4 key reasons why this paper should potentially be rejected from the conference. Again, for each reason, use sub-bullet points to provide detailed arguments and evidence from the paper to support that reason. After weighing the reasons for and against, think of some questions you have about the work.

Then, finish the "Clarity, Quality, Novelty And Reproducibility" and "Summary Of The Review" sections.

Remember, as a reviewer your job is to rigorously evaluate the strengths and weaknesses of the work and to provide critical but constructive feedback to the authors. Be thorough, specific and detailed in your arguments and feedback. Highlight both the positives and negatives you see, and justify your points carefully with reference to the content of the paper.

**User Prompt**

{Abstract of the Paper}

## D.4 MANIPULATIONS

### D.4.1 REVIEW MANIPULATION: LLM REPHRASING

To rephrase the review without altering the semantics, we employ GPT-4o and Llama-3.1-instruct to rewrite the original review judgments, modifying the phrasing while maintaining the original meaning. The same prompt is used for both models to accomplish this task.

**System Prompt**

You are tasked with rewriting a peer review evaluation. You should follow these guidelines:

1. Maintain the overall structure and organization of the review.

2. Improve the writing and make the language more natural and native-sounding.

3. Correct any grammatical errors or awkward phrasing.

Remember to maintain the overall structure and content of the original review, but aim to enhance its readability and fluency.

**User Prompt**

```
{Original Review Judgments}
```

### D.4.2 REVIEW MANIPULATION: MEANINGLESS ELONGATION

We employ a verbose and uninformative template content alongside summaries of the existing review to achieve a meaningless elongation of the original review judgments. Specifically, we adhere to the fixed review format established by ICLR 2023, which consists of four sections. At the beginning of each content section within these sections, we incorporate an identical introductory statement. Additionally, we utilize GPT-4o to generate a one-sentence summary for the "Strengths and Weaknesses" section of the review, embedding this summary within the added text. Below is an example, including the meaningless content we add to the original review.

**Before Manipulation**

Summary Of The Paper:

This is the first sentence. This is the second sentence. This is the third sentence.

Strengths And Weaknesses:

This is the first sentence. This is the second sentence. This is the third sentence. This is the fourth sentence.

Clarity, Quality, Novelty And Reproducibility:

This is the first sentence.

Summary Of The Review:

This is the first sentence. This is the second sentence.

**After Manipulation**

Summary Of The Paper:

This section provides an overview of the key contributions, methodologies, and findings presented in the paper. It summarizes the main arguments and highlights the scope and significance of the research conducted. Specifically, This is the first sentence. This is the second sentence. This is the third sentence.

Strengths And Weaknesses:

In this section, I will discuss the strengths and weaknesses of the paper, focusing on its contributions to the field, methodological rigor, and areas where improvements could be made. The aim is to provide constructive feedback that can help enhance the quality of the work. This paper has several notable strengths, including {Strength Summary}. However, there are some areas that could be improved, such as {Weakness Summary}. Specifically, I discuss the strengths and weaknesses of the paper as following.

This is the first sentence. This is the second sentence. This is the third sentence. This is the fourth sentence.

Clarity, Quality, Novelty And Reproducibility:

I will evaluate the clarity of the paper's presentation, the quality of the research, the novelty of the findings, and the reproducibility of the experiments. This assessment will address whether the research is presented in a clear and coherent manner, offers new insights, and can be replicated based on the provided information. This is the first sentence.

Summary Of The Review:

This summary encapsulates the main points of the review, reflecting on the paper's contributions, strengths, and areas for improvement. It provides a balanced overview of the paper's impact and offers final recommendations for the authors. This is the first sentence. This is the second sentence.

The prompts that request GPT-4o to generate the Strength Summary and Weakness Summary are presented as follows.

**System Prompt**

You are given a peer review of a scientific paper, please identify two key {"strengths" or "weaknesses"} of the scientific paper from the review in a single, concise line, using two phrases separated by 'and'.

**User Prompt**

```
{Original Review Judgments}
```

## E    GEM / GEM-S WITH FINE-TUNED EVALUATION-LM

We not only use the original Llama-3.1-8B-Instruct as the Evaluation-LM, but also fine-tune it to to better estimate the conditional and marginal probabilities. In this section, we provide the details of how we fine-tune the model, as well as the experimental results.

### E.1    IMPLEMENTATION

We fine-tune the original Llama-3.1-8B-Instruct model using 1,000 papers and their corresponding reviews from ICLR 2023 (which are separate from the papers used for experiments). Let gen(metric,$x$,$y$) represent the function that generates the full prompt for predicting judgments (both input and output), where $x = $Null indicates that we are estimating the marginal probability. For each paper, we randomly selected three reviews, denoted as $r_1$, $r_2$, and $r_3$, along with their preprocessed versions $\hat{r}_1$, $\hat{r}_2$, and $\hat{r}_3$. From each paper, we generated nine data points for GEM and nine data points for GEM-S as follows: gen(GEM(-S),$\hat{r}_1$,$\hat{r}_2$), gen(GEM(-S),$\hat{r}_1$,$\hat{r}_3$), gen(GEM(-S),$\hat{r}_2$,$\hat{r}_1$), gen(GEM(-S),$\hat{r}_2$,$\hat{r}_3$), gen(GEM(-S),$\hat{r}_3$,$\hat{r}_1$), gen(GEM(-S),$\hat{r}_3$,$\hat{r}_2$), gen(GEM(-S),Null,$\hat{r}_1$), gen(GEM(-S),Null,$\hat{r}_2$), gen(GEM(-S),Null,$\hat{r}_3$).

In total, this process yield 18,000 input-output pairs. We used the Unsloth framework to fine-tune the model, running on a physical server equipped with a single NVIDIA A100 GPU, provided by Google Colab. Here is the settings we use for fine-tuning:

```
"model_config": {
    "base_model":"unsloth/Meta-Llama-3.1-8B-Instruct-bnb-4bit", # The base model
    "max_seq_length": 4096, # The maximum sequence length
    "load_in_4bit": True, # Load the model in 4-bit
},
"lora_config": {
    "r": 16, # The number of LoRA layers 8, 16, 32, 64
    "lora_alpha":16, # The alpha value for LoRA
    "lora_dropout":0, # The dropout value for LoRA
},
"training_config": {
    "per_device_train_batch_size": 2, # The batch size
    "gradient_accumulation_steps": 4, # The gradient accumulation steps
    "warmup_steps": 5, # The warmup steps
    "max_steps":0, # The maximum steps
    "num_train_epochs": 3, # The number of training epochs
    "learning_rate": 2e-4, # The learning rate
    "optim" :"adamw_8bit", # The optimizer
    "weight_decay" : 0.01,  # The weight decay
    "lr_scheduler_type": "linear", # The learning rate scheduler
    "seed" : 42, # The seed
}
```

### E.2    EXPERIMENTAL RESULTS

| Evaluation Metric | Sentence Deletion | Deletion & Completion | Abstract-only Review |
|---|---|---|---|
| GEM-raw | **-0.123** | **-0.126** | 0.020 |
| | (-0.126,-0.120) | (-0.132,-0.121) | (0.017,0.023) |
| GEM | **-0.401** | **-0.308** | **-0.191** |
| | (-0.448,-0.354) | (-0.358,-0.258) | (-0.261,-0.122) |
| GEM-S | **-0.409** | **-0.206** | **-0.566** |
| | (-0.455,-0.362) | (-0.254,-0.158) | (-0.639,-0.492) |
| GEM-finetune | **-0.155** | **-0.135** | -0.043 |
| | (-0.199,-0.110) | (-0.180,-0.090) | (-0.106,0.020) |
| GEM-S-finetune | **-0.192** | **-0.099** | **-0.355** |
| | (-0.244,-0.140) | (-0.148,-0.049) | (-0.422,-0.288) |

Table 13: Standardized Mean Differences (SMD) $d := (\mu' - \mu)/\sigma_{\text{pooled}}$ of scores after degradations of various evaluation metrics with 95% confidence intervals in parentheses. Significant score decreases ($p < 0.05$) after degradations are highlighted in bold green, implying the evaluation metric can **effectively** penalize the degradation.

| Evaluation Metric | GPT-4o Rephrase | Llama3.1 Rephrase | Meaningless Elongation |
|---|---|---|---|
| GEM-raw | -0.123 | -0.126 | **0.020** |
| | (-0.126,-0.120) | (-0.132,-0.121) | (0.017,0.023) |
| GEM | -0.058 | -0.107 | -0.063 |
| | (-0.090,-0.026) | (-0.143,-0.070) | (-0.097,-0.030) |
| GEM-S | -0.046 | -0.114 | -0.070 |
| | (-0.079,-0.013) | (-0.149,-0.078) | (-0.104,-0.036) |
| GEM-finetune | -0.014 | -0.010 | -0.056 |
| | (-0.044,0.015) | (-0.044,0.024) | (-0.088,-0.025) |
| GEM-S-finetune | -0.026 | -0.045 | -0.074 |
| | (-0.062,0.010) | (-0.085,-0.004) | (-0.112,-0.037) |

Table 14: Standardized Mean Differences (SMD) of scores $d := (\mu' - \mu)/\sigma_{\text{pooled}}$ after manipulations of various evaluation metrics with 95% confidence intervals in parentheses. Significant score increase ($p < 0.05$) after manipulations are highlighted in bold red, implying the evaluation metric are **not robust** against the manipulation.

### E.3 GRE-BENCH BASED ON FINETUNED GEM AND GEM-S

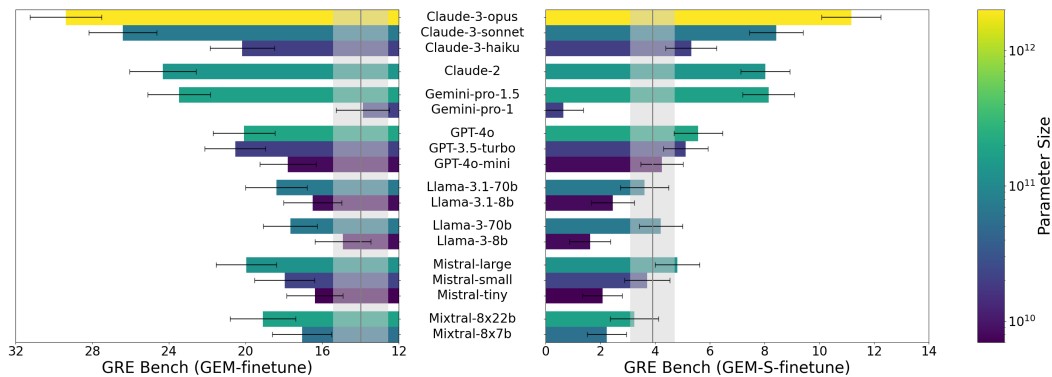

Figure 5: Results of GRE-bench based on finetuned GEM and GEM-S with 90% confidence intervals vs. model parameter size. The grey line represents the average human baseline, with the 90% confidence interval shaded in grey.

