# OpenReview forum: "Benchmarking LLMs' Judgments with No Gold Standard"
_ICLR.cc/2025/Conference — ICLR 2025 Poster_

### Official Review · Reviewer_jK45 · 2024-10-26

**Soundness:** 2
**Presentation:** 3
**Contribution:** 3
**Rating:** 6
**Confidence:** 3

**Summary:**

This work is motivated by the importance of finding a benchmarking method that is not reliant on gold-standard references. The proposed method(s) also attempt to circumvent known issues about using generative models as evaluators, where they are often sensitive to degradation strategies and score-increasing manipulations. The method is empirically compared against a wide range of widely known evaluation metrics to showcase its effectiveness.

The authors also introduce a new benchmark for review generation which is built upon their method that inherits all its benefits. The benchmark is also claimed to combat data contamination by being updated with new review data as it comes out.

**Strengths:**

The paper presents a valuable step towards doing away with the reliance on gold-standard references in evaluation. It's original and creative in the way it makes use of mutual information in the context of generative models.

The method is described rigorously and thoroughly. The flow and structure of the paper makes sense and conveys their ideas well.

The authors also seem to be highly aware and open about the limitations of the method. The paper actively critiques itself and presents answers and discussions on these critiques.

**Weaknesses:**

While the paper is generally highly verbose discussion-wise, I find the experiments and results parts of the paper to be a bit lacking in discussion and analysis. My concerns on this matter are made more specific in the "Questions" part of this review.

Some minor grammar/spelling mistakes: lines 96-97 "..the score can relies solely...", line 455 "Table" is misspelled.

**Questions:**

1. I'm not sure I understand the setup of the first result, am I right in understanding that what you do here is generate k candidates per review, and then get the average GEM score for each candidate by averaging over all other generated candidates when used as references? And then you make the classification by choosing the candidate that resulted in the highest GEM score? How is the classification made? If I'm correct in understanding this, then I don't see how the following sentence from the conclusion applies: "...while our method does not require gold-standard references, it requires reference that includes original human-generated information." Since the way I understand it now, it doesn't really need any human-generated references to generate these scores.

2. Why do you think LMExaminer performs so much better in Sentence Deletion and Deletion & Completion? I feel like that's quite a significant result that hasn't been discussed.

3. "Figure 5 shows that, within each LLM family, such as the Llama-3 or Claude series, models with larger parameter sizes tend to perform better. Moreover, newer versions tend to outperform their predecessors with similar model sizes, as seen in comparisons between Llama-3.1 and Llama-3. These observations further validate the effectiveness of our GEM and GEM-S metrics in accurately evaluating LLMs."
Can GRE-Bench only work with the GEM metrics? Do you think you would observe something similar if you used different metrics?

---

> ### Author Response · Authors · 2024-11-20
>
> **Q1.** This is an interesting question. You are correct that in result one (Section 4.1), there are k human reviews for each project proposal in the instructor-annotated peer grading dataset, and we get the average GEM score for each candidate. We choose human-generated references over LLM-generated ones because the incentive mechanism relies on the assumption of conditional independence among reviews. This means each review should be independent of others, given a specific project proposal being reviewed.
>
> However, this conditional independence assumption may not hold for LLM-generated reviews, especially those from the same LLM family. These reviews often exhibit similarities in their generation process, leading to potential correlations. Using correlated LLM-generated reviews as references can bias the GEM scores, potentially leading to inflated evaluations, especially for models from the same family.
>
> In conclusion, while our method does not require gold-standard references in the traditional sense, it necessitates human-generated information to establish a conditional independent baseline.
>
> However, it's important to acknowledge that if human reviewers were to solely rely on AI-generated content, the quality and diversity of human-generated information could diminish, potentially impacting the effectiveness of our approach, as we discussed in Section 6. This also echoes recent research demonstrating that the absence of original human-generated data in AI evaluation can result in model collapse.
>
> **Q2.** We hypothesize LMExaminer is sensitive to the presentation quality even if it is prompted to evaluate the semantic quality following Review Quality Indicators (RQIs) including four aspects, understanding, coverage, substantiation, and constructiveness. (More details are provided in Section 4 and Appendix E.4.) This hypothesis can be supported by the evidence that LMExaminer significantly increased scores after the meaningless elongation and GPT rephrase.
>
> In Sentence Deletion degradation, since we delete every other sentence, the logical flow can be fragmented. And even if we complete the deleted sentences, the presentation quality may still be worse than the original.
>
> Thank you for pointing out this and we will add this discussion in the revised version.
>
>
> **Q3.** We can expect similar results with metrics that are accurate, such as LMExaminer which also shows a positive correlation with human annotation, while other metrics, such as BERTScore, BLEU, and ROUGE, may not have similar results.
>
> However, a good benchmark should be manipulation resistant, otherwise, the LLM developers may gain an unfair advantage in the benchmark, for example, as the LMExaminer is not robust against meaningless elongation, an unfairly high score may be achieved by training or finetuning models to output unnecessary long responses if we use LMExaminer in GRE-bench. Moreover, LMExaminer needs to take information about the task as input, which can be extremely expensive for peer review, considering the length of the papers.
>
> Still, we believe that using peer review as an evaluation task can lead to a promising field, thanks to the continuous influx of rich data that can circumvent data contamination, and the comprehensive ability required to write a peer review. More metrics that are suitable for evaluating judgment quality should be further explored and studied.
>
> Again, we thank you for your thorough questions, especially in the discussion and analysis of our results. We believe these revisions have substantially improved the paper's rigor and soundness. If any aspects of our responses require further clarification or if additional questions arise, we welcome further discussion to ensure the highest quality of our paper.

---

> > ### Comment · Reviewer_jK45 · 2024-12-02
> >
> > I thank the authors for this extra analysis and their extensive answers. I have increased my score to reflect this.

---

> > > ### Author Response · Authors · 2024-12-04
> > >
> > > Thank you for your thoughtful consideration. We truly appreciate your engagement and are glad our additional analysis and responses were helpful.

---

### Official Review · Reviewer_bCjd · 2024-11-02

**Soundness:** 2
**Presentation:** 3
**Contribution:** 3
**Rating:** 6
**Confidence:** 4

**Summary:**

The paper mainly discusses benchmarking LLM's judgments in the lack of gold standard reference answer, but with a pool of vanilla reference answers by estimating the mutual information between a candidate and reference. The authors also show that this technique is also robust against perturbations and manipulations. Lastly, they also present a benchmark/test set to evaluate how good LLMs can generate high-quality reviews for academic paper.

**Strengths:**

- The paper is well-written, and the problem statements is well-motivated. The authors provide cogent and coherent arguments throughout the paper. I personally think this is a very valid issue for LLM-based evaluators as not always we have a single human-verified gold-standard reference, but a set of good-enough responses to aggregate and use as a ground-truth. The paper also has a good mathematical rigor to prove its hypothesis.
- The benchmark created in the dataset is a plus point as it can help future works.
- I also appreciate the finding that GEM is robust to perturbations as lately there has been a good discussion around LLM having blind spots in evaluation and missing out on subtle disturbances to the input.

**Weaknesses:**

- I do not see any blaring errors in the paper, however, I am a suspicious about the experimental setup as such. Bottom line, the problem statement discusses a **very challenging** problem: _LLMs generating "judicious" reviews for very long and complex scientific papers_. I am not sure if most of the existing LLMs are good enough for such a humongous task and if they do provide good results which can be analyses with some guarantee, as the LLMs are required to have a decent knowledge of the domain as well as a good understanding of the hypothesis in the paper, which even humans struggle to understand in some cases. In a way, this feels like just applying an LLM to a task to see how it fares. But the authors to talk about previous works that have discussed using LLMs as reviewers, and I see that most of them have shown negative results in such cases.
- Related to the above point, I have several questions for the authors, which I hope will be answered to give me some more clarity on the nitty-gritty details of the work.

**Questions:**

- Missing citations:
    - [Finding Blind Spots in Evaluator LLMs with Interpretable Checklists](https://arxiv.org/abs/2406.13439)
    - [Are Large Language Model-based Evaluators the Solution to Scaling Up Multilingual Evaluation?](https://aclanthology.org/2024.findings-eacl.71/)
- In Appendix D3, I see that the whole paper is given in text format, but most of the scientific literature also contains images that are quite important and referenced multiple times. What happens in that case? Similarly, how are tables being managed? Also, are these papers converted from PDF to text format, if so, are they being verified for OCR errors? The papers presented are usually 10 pages in ICLR and 8 pages in *CL, so are you sure the paper is not beyond the token limit of the LLM? Lastly, which LLM was used for the prompt in D3?
- For ICLR dataset, it is mentioned that one reviewer is the candidate and the remaining 2 are reference. Aren't only 2 references too less? I feel in this case the sample bias will be quite high. I urge the authors to clear this for me.
- For the rephrase manipulation, were any sanity checks conducted to make sure that the semantics did not change? Since we are operating at very long-context lengths (document-level), this could be an issue for smaller models like llama-3.1-8b. GPT-4o is quite large and powerful as compared to llama-3-8b and has much better instruction following capabilities, so was this disparity maintained purposefully? If not, I feel llama-3.1-405B or any of the other larger models should be used in place of it which are good at instruction-following.
- For the GRE-Bench, where the LLM is generating reviews, were all 3 human reviews taken as references?
- Line 472 is very redundant and but obvious. Smaller models do not have good instruction following abilities.
- The word Table has been misspelled as Tabel at a couple of places. Please fix it.
- In Figure-4 (and Table-6), I am quite dubious about the parameter counts being mentioned there, as most of closed-source models do not reveal anything, especially for GPT models. Also, the color-scheme in Fig-4 is not comfortable to the eye to make clear demarcations across the bars. Please fix that as well.
- In Prompt D2, the model is provided with one review, which resembles a 1-shot example in a way. In case of generative tasks, there have been cases when the model tries to follow the verbatim or structure of the provided few shots, and the final result lacks creativity. Was this checked in any way?
- What were the generation parameters in each prompting case?
- How were the outputs checked for hallucinations as the generations are quite long?
- In F1, was the same prompt format as in D2 used while finetuning the model?
- There should be some comparison of llama-3.1-8b-Instruct and its finetuned version (as mentioned in F1) to see the gains due to finetuning. Also, finetuning an already finetuned model is slightly tricky as the distribution shift may cause the model to collapse. The general trend is to use a base model directly for this case.

---

> ### Author Response · Authors · 2024-11-20
>
> **W1.** Thank you for raising this important concern. First, we acknowledge that peer-reviewing scientific papers is a challenging task that encompasses multiple levels of abilities. Our approach deliberately leverages this hierarchical structure to create a meaningful evaluation framework.
>
> The review process requires a hierarchy of capabilities, ranging from lower-level abilities like information retrieval and summarization, to mid-level abilities such as domain-specific knowledge application and methodology assessment, and ultimately to higher-level abilities including critical analysis, hypothesis evaluation, and scientific merit judgment.
> We acknowledge the previous negative results in using LLMs to fully automate peer review, however, recent work [Liu and Shah, 2023; Shah, 2024] has shown that LLMs can achieve competitive-to-human-reviewer and sometimes even better effectiveness when testing their mid-level capabilities to detect technical mistakes in peer review. For example, in an experiment by Shah [2024], GPT-4 finds about half of deliberately inserted errors in papers, in contrast, only 1 out of 79 human researchers find one error.
>
> Notably, rather than claiming LLMs can fully replace human reviewers, we are using the peer review task to evaluate LLM capabilities at multiple levels. This setup allows us to measure performance across different cognitive hierarchies, where weaker models can be differentiated by their performance on basic tasks like summarization, while stronger models can be evaluated on more complex abilities such as critical analysis. This creates a natural progression for measuring LLM advancement. We leverage these known limitations to create meaningful evaluation metrics that can help us better understand the current state of AI technology.
>
> To make this evaluation more precise, we adjust synopsis granularity to target specific abilities and use GEM-S metrics to evaluate different aspects separately. For example, by conditioning on the abstract, GEM-S (abstract) evaluates the LLM's ability to generate judgments beyond summarizing general topics and keywords, including retrieving author-stated strengths and weaknesses (ASSW) and providing critical feedback. By further conditioning on a summary of these author-stated strengths and weaknesses, GEM-S (ASSW) narrows the focus to the LLM's capacity for critical thinking and delivering constructive feedback.
>
> Through this hierarchical information structure, we can better understand the progression of LLM capabilities in this hierarchy while maintaining a realistic perspective on their current limitations. This understanding is crucial for both the development of AI technology and its appropriate application in academic contexts.
>
> We will add this discussion about hierarchical information structure to our paper in later revisions.
>
> Reference
>
> Shah, N. B. The role of author identities in peer review. Plos one, 18(6), e0286206. 2023.
>
> Liu R, Shah N B. Reviewergpt? an exploratory study on using large language models for paper reviewing. arXiv preprint arXiv:2306.00622, 2023.

---

> > ### Author Response · Authors · 2024-11-20
> >
> > Thank you again for the detailed questions, and we now provide detailed answers as follows:
> >
> > > Q1. Missing citations
> >
> > Thank you for pointing out these relevant works. We will incorporate these citations in a later revision. They provide a more comprehensive background of the limitations of current LLM evaluators, which further emphasizes the urgency of manipulation resistance in evaluation.
> >
> > > Q2.1. In Appendix D3, I see that the whole paper is given in text format, but most of the scientific literature also contains images that are quite important and referenced multiple times. What happens in that case? Similarly, how are tables being managed? Also, are these papers converted from PDF to text format, if so, are they being verified for OCR errors? The papers presented are usually 10 pages in ICLR and 8 pages in *CL, so are you sure the paper is not beyond the token limit of the LLM?
> >
> > We use the sciencebeam PDF parser tool to convert PDFs to text format, following Liang et al.'s [2023] research about LLM-generated peer review. When using the sciencebeam PDF parser, it retains all image and table titles and captions while disregarding other content.
> >
> > While a few ICLR submissions require OCR before parsing into text (5 out of all submissions in 2023), none of the 300 papers we experimented with required OCR, ensuring no OCR errors. Additionally, we have verified that the parsed text of papers in our dataset does not exceed the input token limit of the LLM.
> >
> > While we acknowledge that PDF parsing is an inherently imperfect process that may occasionally introduce artifacts or miss certain elements, our experimental design mitigates potential biases because all LLM models in our evaluation receive identical parsed versions of each paper, and any parsing inconsistencies would affect all models equally, making the relative performance comparisons robust to such noise.
> >
> > It is an interesting future direction to let multimodality LLMs take both text and figures as the input. Since only a few latest open-source models, such as Llama-3.2, have started to support this, we only input the text to make a fair and consistent comparison, but we believe multimodality open-source models will be common in the near future.
> >
> > > Q2.2. Lastly, which LLM was used for the prompt in D3?
> >
> > Regarding the prompt in Appendix D3, as we use generating peer reviews as a task to evaluate LLMs, we use this prompt for all LLMs mentioned in GRE-bench (Generating Review Evaluation Benchmark) in Section 5, including families of Claude, GPT, Llama, etc.
> >
> > > Q3. For ICLR dataset, it is mentioned that one reviewer is the candidate and the remaining 2 are reference. Aren't only 2 references too less? I feel in this case the sample bias will be quite high. I urge the authors to clear this for me.
> >
> > You are correct that sample bias could exist in individual PMI values, especially in subjective tasks, where even human experts may have different opinions sometimes. However, as we stated in the preliminary section, PMI is an unbiased estimator of MI. When we take the average over the dataset containing 300 papers and 600 reference responses, the sample bias of MI is relatively acceptable: we show the confidence intervals in both results of our validation experiments and GRE-bench.
> >
> > > Q4. For the rephrase manipulation, were any sanity checks conducted to make sure that the semantics did not change? Since we are operating at very long-context lengths (document-level), this could be an issue for smaller models like llama-3.1-8b. GPT-4o is quite large and powerful as compared to llama-3-8b and has much better instruction following capabilities, so was this disparity maintained purposefully? If not, I feel llama-3.1-405B or any of the other larger models should be used in place of it which are good at instruction-following.
> >
> > There seems to be a misunderstanding of our paper in this question. The rephrase manipulation is to rephrase the response (the peer review) instead of the whole document (the paper). Here, we aim to test whether the evaluation metrics have a preference for specific language styles in the response.
> >
> > We only give the rephrasing LLM the review, so it can not add any useful information related to the paper to the response. According to the information processing inequality, even if the rephrasing process changes the semantics of the response, as the rephrasing is independent to the paper, the mutual information should not increase. Our experiment results validate that our GEM metric has this property.
> >
> > Furthermore, as previous literature mentioned [Bai, 2024], the GPT examiner has a preference for the GPT-generated language style. However, though our GEM metric uses a Llama-3.1 8B as the evaluation LM to compute conditional probability, it does not have any preference towards the Llama language style. We test this by using both GPT-4o and Llama-3.1 8B to rephrase the reviews, and do not observe a substantial difference, as shown in Section 4.3.

---

> > > ### Author Response · Authors · 2024-11-20
> > >
> > > > Q5.
> > > For the GRE-Bench, where the LLM is generating reviews, were all 3 human reviews taken as references?
> > >
> > > We use the same dataset as sections 4.2 and 4.3, where there are 300 papers, and 2 reviews of each paper are used as references.
> > >
> > > > Q6.
> > > In Figure-4 (and Table-6), I am quite dubious about the parameter counts being mentioned there, as most of closed-source models do not reveal anything, especially for GPT models. Also, the color-scheme in Fig-4 is not comfortable to the eye to make clear demarcations across the bars. Please fix that as well.
> > >
> > > We acknowledge that the parameter sizes of several closed-source models are estimated as described in the Caption of Table 6. You are totally correct that the estimations are not from the official source, we will further emphasize this fact in our paper to improve transparency. Notably, this information is only to provide context, and none of our conclusions or key findings depend on these parameter count estimates.
> > >
> > > > Q7.
> > > In Prompt D2, the model is provided with one review, which resembles a 1-shot example in a way. In case of generative tasks, there have been cases when the model tries to follow the verbatim or structure of the provided few shots, and the final result lacks creativity. Was this checked in any way?
> > >
> > > This is a very interesting question, we acknowledge that the model may not be able to provide a perfect estimation of the conditional distribution over the review judgments, as discussed in Section 6. Instead of directly testing whether the evaluation LM with prompt D2 can provide an accurate estimation of the distribution (which is not clear how to do), we validate whether the whole system (GEM metric) can effectively penalize degradations, correlate with human scores, and be robust against manipulations. Moreover, we also acknowledge the verbatim and structure can influence the LLM’s prediction as “shortcuts”, therefore, in Section 3.2, we propose the preprocessing to standardize the structure and language style, and propose the GEM-S metric to condition out the mutual information from the verbatim in the abstract of the paper. Our results show that GEM and GEM-S have superior performance compared to GEM-raw which does not have preprocessing.
> > >
> > > > Q8.
> > > What were the generation parameters in each prompting case?
> > >
> > > For reproducibility, we use temperature 0 for all API calls, and max_token = 4000. We will include the parameters in our revised paper.
> > >
> > > > Q9.
> > > How were the outputs checked for hallucinations as the generations are quite long?
> > >
> > > We recognize that hallucination may occur in long-form LLM outputs. Our GEM evaluation metric itself can address this issue through several aspects:
> > >
> > > First, our GEM metric is fundamentally based on mutual information, which provides a theoretical foundation for hallucination detection. By the data processing inequality, the introduction of hallucinated content (information not present in the original paper) would necessarily reduce the mutual information between the generated review and the source paper. This means our metric inherently penalizes hallucinated content, as such content represents a divergence from the true information content of the paper being reviewed.
> > >
> > > However, we acknowledge that this theoretical framework raises important questions about the practical detection and measurement of hallucinations in LLM outputs. While our current evaluation framework provides a principled approach through mutual information, the sensitivity of evaluation metrics to different types of hallucinations remains an open research question.
> > >
> > > For instance:
> > > 1. How effectively can different automated evaluation metrics penalize hallucination?
> > > 2. How do different types of hallucinations (e.g., technical details vs. broader claims) affect the reliability of different automated evaluation metrics?
> > >
> > > These questions point to valuable directions for future research in developing more refined evaluation methodologies. We believe our work provides a foundation for exploring these questions systematically, particularly in the context of long-form technical content generation.
> > >
> > > > Q10.
> > > In F1, was the same prompt format as in D2 used while finetuning the model?
> > >
> > > Yes, we use the same prompt format.

---

> > > > ### Author Response · Authors · 2024-11-20
> > > >
> > > > > Q11.
> > > > There should be some comparison of llama-3.1-8b-Instruct and its finetuned version (as mentioned in F1) to see the gains due to finetuning. Also, finetuning an already finetuned model is slightly tricky as the distribution shift may cause the model to collapse. The general trend is to use a base model directly for this case.
> > > >
> > > > We provide the comparison in Table~7 and 8 in Appendix F, the GEM and GEM-S are computed with llama-3.1-8b-Instruct and the GEM-finetune and GEM-S finetune are computed with its finetuned version. The finetuned model does not lead to significantly better results, as we discussed in Section 6.
> > > >
> > > > We appreciate your important point about the finetuning, particularly regarding distribution shift and potential model collapse. Based on your recommendation, we plan to explore finetuning the base model directly in future work to improve our GEM metrics’ performance. While these insights suggest promising future research directions, we believe our contributions of proposing the first, to our best knowledge, accurate and manipulation-resistant evaluation metric with no gold standard reference remain valid and comprehensive.

---

> > > > > ### Author Response · Authors · 2024-11-20
> > > > >
> > > > > In addition, we thank your suggestions on the typos and Figure 4 presentation, we will fix all these in a later revision.
> > > > >
> > > > > Again, we thank you for all the detailed and insightful questions and suggestions, which have helped us strengthen the completeness and presentation of our research. We have carefully addressed each point raised and hope our responses can address your confusion and concerns and provide a better understanding of our paper. Please feel free to let us know if we misunderstood any of your questions or if you have any new questions.

---

> > > > > > ### Comment · Reviewer_bCjd · 2024-12-03
> > > > > > **Final Official Comment to the Authors**
> > > > > >
> > > > > > Dear Authors,
> > > > > >
> > > > > > Thank you for the detailed response and sorry for the delay. I appreciate the detailed and clear rebuttal responses provided and I urge that this information be included in the paper as well. However, I am still unclear about the PDF parsing in Q2. Even when images are retained, a normal LLM cannot parse them. And without it, the paper is a bit incomplete, as there will be missing references. For example, the paper may have a statement like "In Fig X, where we find ABC", and without the image, this statement would be like a dangling pointer. Similarly, a caption might be retained, but the corresponding image would not be in the text when feeding the paper to the LLM.
> > > > > >
> > > > > > I understand these are inevitable problems, and just wanted some more clarification on them. I know this is quite late, and I won't let it affect my judgment. I feel I have adequately scored the paper earlier, and I will retain it for now. Thank you!

---

> > > > > > > ### Author Response · Authors · 2024-12-04
> > > > > > >
> > > > > > > Thank you again for your detailed response and feedback on our paper. We appreciate you taking the time to review the additional information and rebuttal we provided.
> > > > > > >
> > > > > > > Regarding your request to incorporate the rebuttal information into the paper, we have worked to integrate it into the revision. This includes the refined GEM-S metric leveraging the hierarchical information structure of peer review tasks and all experiment details needed for reproducibility, as noted in the Revision Note above. Please take a look and let us know if you have any further questions.
> > > > > > >
> > > > > > > Regarding your question about the pdf parsing, We would like to address your concerns and clarify a few points:
> > > > > > >
> > > > > > > 1. Our main contributions are not significantly influenced by the PDF parsing process for the following reasons:
> > > > > > >
> > > > > > > a) Our first contribution, the proposal of the GEM metric as an accurate and manipulation-robust metric for NLG evaluation without gold standard reference, does not rely on this parsing process. The validation experiments have shown its effectiveness.
> > > > > > >
> > > > > > > b) For our second contribution, the GRE-bench, as discussed in the previous response, our experimental design mitigates potential biases introduced by the parsing process. All LLM models in our evaluation receive identical parsed versions of each paper, and any parsing imperfections would affect all models equally, making the relative performance comparisons robust to such noise.
> > > > > > >
> > > > > > > 2. Despite the limitations of the PDF parsing approach, previous work by Liang et al. [2023] has demonstrated that LLMs can provide helpful comments for papers using this method. We acknowledge that the parsing approach is not perfect, but it is a starting point for exploring the potential of LLMs in generating peer reviews.
> > > > > > >
> > > > > > > 3. As mentioned in our previous response, the limitations of the PDF parsing approach lead to interesting future directions, such as investigating how multimodal models can improve review performance through their ability to read figures. This opens up new avenues for research in this area.
> > > > > > >
> > > > > > > We understand your concern about the effectiveness of using LLMs to fully-automatically generate peer reviews. However, we believe that recent literature [Liu and Shah, 2023, Liang et al., 2023] has already validated the possibility of using LLMs to provide useful feedback on scientific papers. The key challenge lies not in their technical ability to read PDFs but in their lack of critical thinking and domain knowledge compared to human reviewers. As mentioned in the previous response and the revised version of our paper, we have added the fine-grained GEM-S that leverages the hierarchical information structure of peer review to better evaluate LLMs, inspired by your previous comments.
> > > > > > >
> > > > > > > We believe the revised manuscript addresses your main concerns and has been substantially improved by your and all other reviewers’ valuable input. If you agree that we have made progress in the quality and clarity of the work, we would greatly appreciate it if your score could be updated to reflect this.
> > > > > > >
> > > > > > > Thank you again for your constructive and detailed feedback throughout the review process. We believe the revised manuscript addresses your main concerns and has been substantially improved thanks to your and all other reviews’ efforts. If you agree that we have made progress in the quality and clarity of the work, we would greatly appreciate it if you could consider updating your score to reflect all these efforts.
> > > > > > >
> > > > > > > ### Reference
> > > > > > >
> > > > > > > Liu R, Shah N B. Reviewergpt? an exploratory study on using large language models for paper reviewing. arXiv preprint arXiv:2306.00622, 2023.
> > > > > > >
> > > > > > > Liang W, Zhang Y, Cao H, et al. Can large language models provide useful feedback on research papers? A large-scale empirical analysis. arXiv e-prints, arXiv: 2310.01783. 2023.

---

### Official Review · Reviewer_CxHL · 2024-11-03

**Soundness:** 3
**Presentation:** 3
**Contribution:** 3
**Rating:** 6
**Confidence:** 3

**Summary:**

This paper presents GEM (Generative Estimator for Mutual Information), a new evaluation metric for assessing the quality of text responses generated by LLMs in tasks that lack a clear gold standard reference.  GEM aims to provide an accurate and manipulation-resistant metric by estimating the mutual information between candidate responses and a set of reference responses, which are not required to be perfect. The main idea is to measure the semantic informativeness of candidate responses by filtering out aspects like language style and syntax, focusing on how much information the candidate's response reveals about the references.

**Strengths:**

The paper is clearly written and rigorously formulated. The research topic is very timely and important, given the increasing use of LLMs and the challenges of evaluating their performance in subjective tasks.

**Weaknesses:**

I have some questions and concerns:

1. Impact of Suboptimal Human References: I am concerned about GEM's reliability when human references are of lower quality than LLM outputs. Could the LLM used for estimating probability distributions learn to favor responses that resemble flawed human references, even if they are less informative, potentially leading to inaccurate scores?

2. Performance stability according to the choice of evaluation-LM and preprocessing LLM: How sensitive is GEM to the choice of LLMs used for evaluation (Llama-3.1) and preprocessing (GPT-4)? Would using different LLMs, especially open-source ones, impact the results? LLMs are released and updated frequently, and I wonder how we can deal with such frequent updates to make the GEM score consistent and reliable over time.

**Questions:**

This is a kind of optional question out of my curiosity.

It seems that GEM is a relative scoring/evaluation framework. If so, what would be the pros and cons of GEM compared to other relative evaluation frameworks, such as the ELO rating system commonly used in generation task leaderboards, like those in Chatbot arenas?

---

> ### Author Response · Authors · 2024-11-20
>
> **W1.** The advantage of GEM is encouraging participants to provide informative responses, even when the quality of peer responses may vary. By rewarding participants based on the mutual information between their response and their peer's response I(X;Y), the data processing inequality guarantees that the less informative version of X pays less than X, regardless of Y’s quality (see proposition 3.1).
> However, it's important to acknowledge the limitations of Data Processing Inequality. Firstly, it requires Y to be at least somewhat informative. Secondly, the noisy versions of X and Y must be independent conditional on the original X.
>
> Two scenarios can compromise the mechanism:
>
> 1. Independent Y: If Y is independent of everything, including X, noisy X and X become indistinguishable, leading to suboptimal outcomes.
>
> 2. Violated Conditional Independence: When both Y and noisy X are low-quality AI-generated content, this assumption may not hold, potentially rewarding less informative responses unfairly. This also echoes recent research demonstrating that the absence of original human-generated data in AI evaluation can result in model collapse.
>
> Our results indicate that the human references in ICLR2023 are good enough for the GRE bench based on the GEM metric to effectively distinguish different LLMs. To mitigate these issues in the future, using a diverse range of responses can be useful.
>
> **W2.** Per your suggestions, we conduct experiments with various preprocessing models and evaluation LMs. Our findings remain consistent across various preprocessing models and evaluation language models. For more details, please see our general reply above.
>
> **Q1.** Our GEM metric differs from the ELO rating system in the following ways.
>
> 1. As the name, Generative Estimator for Mutual Information, suggests, the GEM metric has the same interpretation of mutual information, i.e. how much information knowing one variable can inform us about the other (in bits).
> 2. The GEM score is static, i.e., we don’t need to match for pair-wise competition, which saves computation power, especially when the number of LLMs goes up.
> 3. The ELO rating system still needs an examiner to pick the winner. Compared to human examiners, our method saves the money of hiring humans, while LM examiners are not robust against manipulations, as our paper and some previous work suggest.
>
> This inspiring question indicates that there could be an interesting future direction to apply our GEM metric to the ELO rating systems, such as the Chatbot arena, as an examiner. This may create arenas in more scenarios that evaluate subjective judgments, such as peer review, book review, and other qualitative content generation tasks.
>
> Thank you again for your insightful questions and suggestions, which help us further improve our completeness and indicate interesting future directions. We hope our responses have addressed your questions and provided you with a better understanding of our research. Please feel free to let us know if you need any further clarification.

---

> > ### Comment · Reviewer_CxHL · 2024-11-22
> >
> > Thank you for the clarification. I updated my score accordingly.

---

> > > ### Author Response · Authors · 2024-11-22
> > >
> > > Thank you for updating your score. We appreciate your time and thoughtful evaluation. Please let us know if you have any further feedback or questions.

---

### Official Review · Reviewer_8Skk · 2024-11-06

**Soundness:** 3
**Presentation:** 3
**Contribution:** 3
**Rating:** 6
**Confidence:** 4

**Summary:**

The paper introduces GEM, a novel metric for evaluating the performance of large language models in generating informative judgments using given references but not necessary gold standards, and presents GRE-bench, a benchmark for assessing the quality of peer reviews generated by LLMs.

GEM is designed to assess the informativeness of judgments in scenarios where gold standard references are unavailable, expanding the scope of benchmarking to subjective tasks like peer review.

**Strengths:**

Using mutual information for evaluation subjective generative response is novel as my understanding.

The paper give theoretic guarantees of the proposed method.

Experiments show the method works well on the task of evaluating reviews for academic papers or student proposals.

**Weaknesses:**

Only one type of evaluation tasks has been assessed: paper review evaluation.  It would be more convicing if the paper can conduct experiments of more subjective evaluation tasks, such as summarization, dialog, etc.

The paper uses Llama 3.1 8B as the evaluation LM.  I would like to see how different LMs effect the performance of the evaluation.

Figure 4 is not mentioned in the paper.

**Questions:**

Using LLM to estimate the conditional probility seems problematic:  While P(x,y)=P(x)*P(y|x)=P(y)*P(x|y), P_LLM(x)*P_LLM(x|y) != P_LLM(y)*P_LLM(x|y).  I wonder if this will cause problems in the estimation.

---

> ### Author Response · Authors · 2024-11-20
>
> **W1.** We appreciate your suggestions on the diversity of evaluation tasks.
>
> Currently, we focus on peer review evaluations for several reasons: first, previous probability-based metrics (BARTScore, GPTScore) have been shown to be effective in traditional NLG evaluation tasks like summarization and dialogue. Our GEM can be regarded as a probability-based evaluation metric like previous metrics, so we can expect similar results in these tasks.
>
> Second, tasks like summarization and dialogue have clear human references as gold standards. However, peer review is inherently subjective, with diverse and sometimes conflicting judgments among human reviewers. This makes it a more complex and nuanced evaluation scenario. As our title suggests, we aim to benchmark LLMs’ ability to generate informative judgment without gold standard reference, peer review is an ideal setting.
>
> In addition, our GEM metric is the base of the GRE-bench (Generating Review Evaluation Benchmark), so it’s important to validate GEM’s accuracy and manipulation resistance on peer review tasks, which also implies the effectiveness of the GRE-bench.
>
> In the long term, we are excited about the potential of GEM to evaluate the quality of book reviews, movie reviews, and other subjective content, provided suitable public datasets become available.
>
> Per your suggestion, we have found an open-access dataset of Yelp restaurant reviews. We are working on an experiment validating the GEM metric in the evaluation of Yelp reviews. Specifically, testing whether the degradation will significantly decrease the GEM scores of Yelp reviews. While we aim to include these results in the camera-ready version, we are not sure whether we can finish that by the end of the rebuttal.
>
> As you mentioned in the strength, we have conducted three experiments on both student proposal review and ICLR review, which provide a variety of expertise and task difficulty, we believe that the current positive results effectively show GEM’s accuracy and manipulation resistance being superior to the baselines including the popular GPT4o examiner. Still, we value your suggestion on task diversity which will further strengthen our method’s soundness.
>
>
> **W2.** We have conducted experiments with different preprocessing models and evaluation LMs, generally, the results remain consistent across models. Please refer to the general response above for the details.
>
>
> **W3.** Line 471, we aim to refer to Figure 4 instead of Figure 5, sorry for the typo, we will fix it in the revised version.
>
>
> **Q1.** Thank you for your insightful question. We have conducted experiments to check whether Pr_LLM[x|y]*Pr_LLM[y] = Pr_LLM[y|x]*Pr_LLM[x], and the results revealed a high positive correlation between the two values, although they were not precisely equal. Despite this difference, the empirical results in our paper indicate that this does not negatively impact the desired properties, including the accuracy and manipulation resistance.
>
>
> Specifically, we randomly select 500 papers from ICLR 2023 and 1874 corresponding reviews in total. For each pair of reviews x,y for the same paper, we use our method to compute log Pr_LLM[x|y], log Pr_LLM[y|x], log Pr_LLM[x], and log Pr_LLM[y]. For those 5328 pairs, we compute the Spearman correlation between Pr_LLM[x|y]*Pr_LLM[y] = exp(log Pr_LLM[x|y] + log Pr_LLM[y]) and Pr_LLM[y|x]*Pr_LLM[x] = exp(log Pr_LLM[y|x] + log Pr_LLM[x]) is 0.943, which indicates that they are highly positively correlated, despite minor differences. Potential noise in the estimation process may account for these differences.
>
> We will include a discussion of this question in our revised version.
>
> Thank you again for your thoughtful feedback, which has helped us further strengthen the completeness of our research. We hope our responses have clarified your concerns and provided a more comprehensive understanding and evaluation of our paper. If any points remain unclear or if you have additional questions, please don't hesitate to let us know.

---

### Author Response · Authors · 2024-11-20

We thank all the reviewers for their thoughtful comments and appreciate that reviewers explicitly state our paper is “novel”, “well-motivated”, “well-written” and “rigorous”.

We want to highlight that, to the best of our knowledge, our paper is the first paper exploring evaluating LLMs with tasks that have no gold-standard references through an information theoretical lens.

Additionally, we thank reviewer bCjd for “appreciating the finding that GEM is robust to perturbations”. The manipulation resistance of our GEM metrics implies its effectiveness surpassing the GPT4 examiner in evaluating judgment qualities. We appreciate the suggestions by reviewer bCjd on the literature studying blind spots of LLMs in evaluation, which further support our findings and results.

We also thank reviewer bCjd for acknowledging our contribution to building the GRE-bench dataset where we use 18 LLMs to generate reviews for 300 papers in ICLR2023. Indeed, we also provide a methodology for utilizing the rich and valuable data from ICLR reviews to help benchmark LLMs while circumventing data contamination. We’d like to give credit to all the reviews and the conference for making our methodology possible.

---

> ### Author Response · Authors · 2024-11-20
>
> A common question raised by reviewer 8Skk and CxHL is about the consistency of our GEM metric over various preprocessing models and evaluation LMs. We appreciate this question and regard it as a great opportunity to improve the soundness of our methods. Here are the research questions:
>
> 1. Does using an open-sourced preprocessing model (rather than GPT4o) still lead to effective GEM metric and similar GRE-bench results?
>
> To answer this question, we use a Llama3.2-90b model for preprocessing and re-run all experiments mentioned in our paper. We find that most experimental results remain valid except for two manipulation tests. The GRE-bench results also remain highly consistent with the results of using GPT4o preprocessing.
>
> Specifically, GEM shows a significant positive correlation with human annotation in the proposal peer grading dataset (Section 4.1), however, the Spearman’s correlation coefficient between GEM scores and instructor grades drops from 0.433 (GPT4o preprocessing) to 0.274 (Llama3.2-90b preprocessing). In the degradation experiment, GEM with Llama3.2-90b preprocessing also shows significant sensitivity to all degradations and even performs better in “sentence deletion” and “deletion and completion” than using GPT4o to preprocess. In the manipulation experiments, it is robust against “GPT4 rephrase” but fails in the “Llama-3.1 rephrase” and “Meaningless Elongation”.
>
> Based on the GEM metric with llama3.2-90b preprocessing, we re-compute the GRE-bench scores. The Spearman's correlation coefficient between the GRE-bench scores based on the GEM metric with Llama3.2-90B preprocessing and the original scores shown in our paper with GPT4o preprocessing is 0.89. The analogous coefficient for the GRE-bench scores based on GEM-S is 0.85. We will provide all detailed results in the revised version.
>
> Overall, since better models can more accurately preprocess the reports, the GEM metric will be more accurate and manipulation-resistant with superior preprocessing. This also echoes the question from reviewer CxHL “LLMs are released and updated frequently, and I wonder how we can deal with such frequent updates to make the GEM score consistent and reliable over time.” Our answer would be that preprocessing with better LLMs can lead to better accuracy and reliability, and our GEM metrics will benefit from the ongoing advancements in LLM technology.
>
>
> 2. Does using a different evaluation LM still lead to consistent results?
>
> We use Llama-3.1 8B and Llama-3.1 70B as the evaluation LM to estimate the pointwise mutual information, and the results are consistent over these two models.
>
> The Spearman's correlation coefficient between the GRE-bench scores based on the GEM metric computed using Llama-3.1 8B and that same score computed with Llama-3.1 70B is 0.93. The analogous coefficient for the GRE-bench scores based on GEM-S is 0.92. We will further provide all scores in the revised version.
>
> This consistency shows that with our metrics, a small model with 8B parameters that can be deployed on a single NVIDIA L4 GPU, plus proper preprocessing, is still effective in evaluating the quality of judgments generated by much larger models, though larger evaluation LM may lead to better evaluation performance.
>
> Moreover, as reviewer CxHL noted, “LLMs are released and updated frequently”, this observation supports the consistency and reliability of GEM score over different models. We will include these results in our paper, and we do appreciate the valuable suggestions from our reviews.
>
> We acknowledge that there are thousands of potential models we could test for preprocessing and evaluation. While it's impossible to exhaustively test every available large language model, we believe our current experimental results demonstrate the solid effectiveness of the GEM metric. These additional experiments, suggested by our reviewers, have further strengthened the rigor of our methodology.

---

### Comment · Area_Chair_DSX5 · 2024-11-21
**Reminder: Please respond and update the score if necessary**

Dear Reviewers,

Kindly ensure that you respond proactively to the authors' replies so we can foster a productive discussion. If necessary, please update your score accordingly. We greatly appreciate the time and effort you’ve dedicated to the review process, and your contributions are key to making this process run smoothly.

Thank you,

AC

---

### Author Response · Authors · 2024-11-28
**Revision Notes**

Dear Reviewers,

Thank you again for your thorough and constructive feedback on our paper!

We are gratified that the reviewers found our paper to be novel, well-motivated, well-written, and rigorous. This positive assessment affirms the significance of the problem we are addressing and the strength of our proposed methodology. At the same time, your insightful questions and critiques have enabled us to further strengthen our paper.

In this revision, we have carefully addressed each of the points raised by the reviewers. Key updates include:

1. Additional experiments validating the robustness of our GEM metrics across various preprocessing models (Appendix A.3) and evaluation models (Appendix A.2), to address questions from reviewers 8Skk and CxHL. Most of our experiment results remain consistent across various preprocessing models and evaluation models.

2. New experiments in Appendix A.4 applying the GEM metric to evaluate Yelp reviews, demonstrating the broad applicability of our approach beyond peer review and peer grading contexts, as requested by reviewer 8Skk.

3. Additional experiments examining the symmetric property of LLM estimated distribution in Appendix A.5, as suggested by reviewer 8Skk.

4. A discussion in Sections 3.2 and 5 of how our GEM-S metric enables hierarchical evaluation across different cognitive levels. By varying the synopsis granularity, GEM-S can differentiate weaker models based on basic abilities like summarization, and stronger models on complex skills like critical analysis. We leverage the nature that peer review is a challenging task that encompasses multiple levels of abilities, as reviewer bCjd suggests, to create a meaningful evaluation framework.

5. Expanded results analysis in Section 4.2, including an explanation for LMExaminer's strong performance on "Sentence Deletion" degradations, as requested by reviewer jK45.

6. Clarifications on the experimental setup, such as API parameters and PDF parsing, added to Appendix C, as requested by reviewer bCjd.

7. Improved presentation of Figure 4, with clearer notation that parameter counts for some closed-source models are estimates, based on reviewer bCjd's feedback.

8. Additional citations added to the Introduction, as suggested by reviewer bCjd.

The full set of major changes are highlighted in **blue** (if the whole section is new, we only highlight the title for better reading experience.) throughout the revised manuscript. We hope these modifications fully address the reviewers' concerns and further strengthen the contributions of our work.

Once again, we are deeply grateful to the reviewers for their constructive critique. We also want to express our appreciation to ICLR for generously providing the review data that made this work possible. We believe our proposed GEM metrics and GRE-bench provide methodologies to make effective use of this invaluable resource to enable more **accurate**, **manipulation-resistant**, and **data-contamination-robust** evaluation of language models on open-ended generation tasks even without gold-standard reference.

We look forward to further discussion and to the opportunity to continue refining our work with your guidance.

---

### Meta-Review · Area_Chair_DSX5 · 2024-12-22

**Metareview:**

The paper introduces GEM, a metric for evaluating large language models (LLMs) based on the informativeness of their generated responses without needing gold-standard references. GEM estimates mutual information between candidate and reference responses, making it ideal for subjective tasks like peer reviews. The accompanying benchmark, GRE-bench, assesses peer review quality generated by LLMs using GEM, offering a robust, manipulation-resistant evaluation. GRE-bench mitigates generative model evaluation issues and updates its data to prevent contamination. GEM's effectiveness is validated against existing metrics.

On a positive note, the reviewers commend the paper for its innovative use of mutual information to evaluate subjective generative responses. The approach provides theoretical guarantees and effectively demonstrates its capability in tasks such as academic paper review evaluation. The paper is clearly written and rigorously structured, effectively addressing the pressing challenge of assessing LLM performance in subjective tasks where gold-standard references are absent. Its mathematical rigor, coupled with the introduction of a new benchmark dataset, significantly contributes to ongoing and future research efforts. Additionally, GEM stands out for its robustness to input perturbations. The authors are also praised for their transparency in acknowledging the method's limitations, offering insightful self-critical analysis and discussions throughout the paper.

The reviewers express several critiques and concerns regarding the paper's scope and methodology. They recommend broadening the evaluation to encompass additional tasks such as summarization and dialogue, which would enhance the study's credibility. The reliance on a single language model, Llama 3.1 8B, raises questions about the potential impact of using diverse models on the results. Moreover, there are concerns about GEM's reliability when faced with suboptimal human references and its sensitivity to the choice of both evaluation and preprocessing LLMs. The capability of current LLMs to produce insightful reviews for complex scientific papers is questioned, suggesting that the study may lean more towards an exploratory approach. Although the paper offers detailed discussions, the sections on experiments and results are perceived as lacking in-depth analysis, prompting the reviewer to seek further clarification in future revisions.

The authors have made significant updates to the paper, effectively addressing the previously raised issues.

The paper makes a significant contribution by introducing a method to estimate mutual information between candidate and reference responses without necessitating the reference to be a gold standard. I think this approach holds the potential for substantial impact. I am leaning to recommend acceptance of the paper.

**Additional Comments On Reviewer Discussion:**

Reviewer 8Skk commends the paper for its novel use of mutual information in evaluating subjective generative responses, backed by theoretical guarantees and strong experimental results for tasks like academic paper reviews. However, they suggest expanding the evaluation to more diverse tasks, like summarization and dialogue, and exploring the effects of different language models, as the current use of Llama 3.1 8B limits insights. They also note that Figure 4 is not referenced within the text.

Reviewer CxHL appreciates the paper's clear writing and timely exploration of evaluating LLM performance in subjective tasks. However, concerns are raised about GEM's reliability if human references are flawed and how sensitive GEM might be to the choice of LLMs, considering frequent updates in LLM technology.

Reviewer bCjd highlights the paper's well-written and motivated problem statement, acknowledging its mathematical rigor and the utility of the benchmark dataset. They appreciate GEM's robustness to perturbations but question the LLMs' capability in generating "judicious" reviews for complex scientific papers, as this might be challenging even for humans.

Reviewer jK45 highlights the innovative approach of using mutual information and notes the paper's rigorous methodology and self-critical perspective on its limitations. Yet, they feel the experiments and results sections could benefit from more detailed discussion and analysis.

The authors addressed the concerns and update the paper accordingly.

---

### Decision · Program_Chairs · 2025-01-22

Accept (Poster)